# Similarity as Reward Alignment: Robust and Versatile Preference-based Reinforcement Learning

## Abstract

Preference-based Reinforcement Learning (PbRL) entails a variety of approaches for aligning models with human intent to alleviate the burden of reward engineering. However, most previous PbRL work has not investigated the robustness to labeler errors, inevitable with labelers who are non-experts or operate under time constraints. We introduce Similarity as Reward Alignment (SARA), a simple contrastive framework that is both resilient to noisy labels and adaptable to diverse feedback formats. SARA learns a latent representation of preferred samples and computes rewards as similarities to the learned latent. On preference data with varying realistic noise rates, we demonstrate strong and consistent performance on continuous control offline RL benchmarks, while baselines often degrade severely with noise. We further demonstrate SARA's versatility in applications such as cross-task preference transfer and reward shaping in online learning.

## 1 Introduction

Reinforcement Learning (RL) algorithms rely on carefully engineered reward functions in order to produce the desired behaviors for a task of interest (Sutton & Barto, 2018; Dann et al., 2023). In complex real-world settings, reward engineering requires various sensors, such as motion trackers (Bin Peng et al., 2020) or computer visions systems (Devin et al., 2018), as well as tedious hand-crafting to fine-tune such functions (Zhu et al., 2020) and ensure safe behavior (Kim et al., 2023). To mitigate reward engineering challenges, Preference-based RL (PbRL) algorithms have garnered increased attention in recent years. In a PbRL setting, human labelers provide feedback on a dataset of agent behaviors, and the PbRL algorithms aim to learn agent models that produce behavior better aligned to the preferences. Prominent examples include Large Language Model (LLM) fine-tuning (Ziegler et al., 2020; OpenAI et al., 2024; Ouyang et al., 2022; DeepSeek-AI et al., 2025) as well as robotics and simulated control (Sadigh et al., 2017; Christiano et al., 2017).

PbRL methods can learn a reward function from human feedback to use in downstream RL, but they face the challenge of accurately representing preferences from limited data (Wirth et al., 2017). Many prior works leverage preference labels on trajectory pairs by applying the Bradley-Terry (BT) model (Bradley & Terry, 1952):

$$P[\sigma^1 \succ \sigma^0; \psi] = \frac{\exp\left(\sum_t \hat{r}(s_t^1, a_t^1; \psi)\right)}{\exp\left(\sum_t \hat{r}(s_t^1, a_t^1; \psi)\right) + \exp\left(\sum_t \hat{r}(s_t^0, a_t^0; \psi)\right)}$$

where $\sigma^1$ and $\sigma^0$ are sampled preferred and non-preferred trajectories, respectively, and $\hat{r}_\psi$ is a learnable reward function. The BT model is often used to learn an explicit reward function $\hat{r}_\psi$ (Christiano et al., 2017; Lee et al., 2021b; III & Sadigh, 2022; Ouyang et al., 2022; Kim et al., 2023) or re-formulated to learn a policy without a reward model (Hejna et al., 2024; Hejna & Sadigh, 2023; An et al., 2023; Kang et al., 2023; Rafailov et al., 2023; Kuhar et al., 2023).

In both cases, the BT model formulations come with assumptions and limitations, discussed by previous works (Sun et al., 2025; Tang et al., 2024; Munos et al., 2024; Azar et al., 2024; Ye et al., 2024). The BT model assumes that human preferences are transitive, an assumption which has been undermined by psychology research (Ye et al., 2024; Tversky, 1969; May, 1954). Azar et al. (2024)

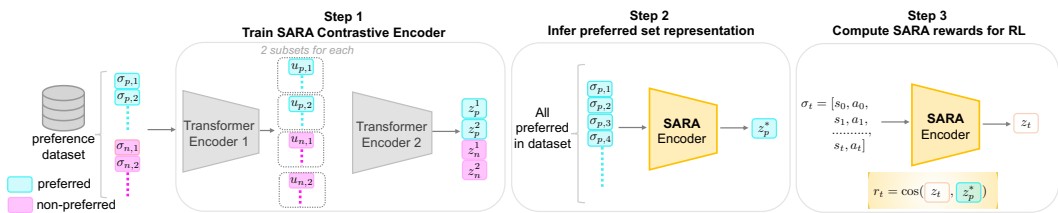

Figure 1: **SARA framework:** Step 1 (training): preferred and non-preferred trajectories ($\boldsymbol{\sigma}_{p,i}$ and $\boldsymbol{\sigma}_{n,i}$) are extracted from preference data. The first Transformer encodes each trajectory into a single token to give us representations $\mathbf{u}_{p,i}$ and $\mathbf{u}_{n,i}$. We then divide the set of preferred $\{\mathbf{u}_{p,i}\}$ into 2 subsets (and likewise for $\{\mathbf{u}_{n,i}\}$). The second Transformer allows attention between the $\mathbf{u}_i$ of the same subset. It encodes each subset into a single latent, so we then have $\mathbf{z}_p^1, \mathbf{z}_p^2, \mathbf{z}_n^1, \mathbf{z}_n^2$. The SimCLR contrastive loss pushes together the two $\mathbf{z}_p^k$ latents, pushes together the two $\mathbf{z}_n^k$, and pushes apart the $\mathbf{z}_p^k$ from $\mathbf{z}_n^k$. Step 2 (infer): Pass all preferred trajectories through SARA encoder to get $\mathbf{z}_p^*$, a fixed representation of preference. Step 3 (RL rewards): in RL training, sample trajectory $\boldsymbol{\sigma}_t$. Compute reward for $\boldsymbol{\sigma}_t$ as cosine similarity to preferred latent, $\mathbf{z}_p^*$.

showed that the BT reward models can overfit to the relative rankings in the trajectory pairs, resulting in agent behavior that also overfits to the preferred ranked trajectory. Overfitting is particularly problematic when labels are noisy or behaviors are similar. Labeling errors occur when annotators are time-constrained or non-experts (Ye et al., 2024; Cheng et al., 2024). Realistic error rates are between 5-38% (Sun et al., 2025), as evidenced by an observed 25% disagreement rate among labelers (Dubois et al., 2023; Coste et al., 2024). Prior work demonstrated that even a 10% label error rate can significantly degrade RL performance (Lee et al., 2021a; Cheng et al., 2024). Sun et al. (2025) show that the BT-model is not a necessary choice for a reward modeling approach, and BT-based models result in underperforming behaviors when labeling error rates are above 10%.

In contrast to most previous work, we assume the presence of labeling mistakes and similar behaviors in ranked pairs, so we avoid learning BT-modeled rewards based on the relative labels. To this end, we introduce Similarity as Reward Alignment (SARA), a robust and flexible PbRL framework (see Figure 1 for an overview). SARA acknowledges that even with noise, discerning patterns exist in the preferred set as a whole and employs contrastive learning to obtain a representation for this set. SARA then computes rewards at each timestep based on the encoded trajectory's similarity to the representation of the preferred trajectories. Despite its simplicity, it handles noisy or ambiguous preference data reliably, and to our knowledge, our framework is novel in the PbRL literature. Our contributions and findings are:

**Strong performance and robustness.** Compared to state-of-the-art baselines, SARA achieves competitive or superior performance using human-labeled preference datasets. We vary the preference data by injecting label noise (0%, 10%, 20%, 40% error rates). SARA results in consistent policy evaluation returns from 0-20% error rates, whereas baseline models fluctuate up to 73%. At the largest error rate 40%, SARA's policy returns degrade but still outperforms or is on par with baselines on most datasets examined. Moreover, we demonstrate that SARA inferred transition rewards correlate better with the environmental transition rewards (unknown at training time) compared to SOTA reward based methods. This is true at all error rates, indicating SARA's robustness to learning preference patterns. We also experiment on human preference data with equally preferred pairs omitted and script labeled preference data, in which preference labels are based on true environment rewards. We again show consistent policy evaluations rewards compared to baselines.

**Versatile preference modeling.** Though our primary focus is a robust PbRL framework, we can further leverage our transition rewards in underexplored applications. We show we can transfer preferences from one locomotion dataset (hopper) to another (walker2d). We also conduct reward shaping in online RL using a cherry-picked preference set rather than ranked pairs. BT based reward models could in principle be used for these applications too, but most works do not apply their models in these unique ways. BT based models also require ranked pairs, so they are not naturally applicable to the feedback format of a cherry-picked preference set. The methods that do not learn an explicit reward network, as in (Kang et al., 2023; Hejna et al., 2024; An et al., 2023;

Kuhar et al., 2023), typically can not provide reward values for sampled trajectories. Therefore, our versatility contributions are the following: 1) our demonstrated effectiveness in using SARA rewards for applications beyond offline RL benchmarks, and 2) our ability to work with different feedback formats as demonstrated in the online RL experiment.

In summary, SARA shows more stable performance compared to baselines using preference datasets with varying error rates. SARA inferred rewards correlate better with environmental rewards. Beyond robustness, SARA offers versatility by enabling cross-task preference transfer and reward shaping in online RL, while not requiring the restrictive paired feedback format for BT models.

## 2 RELATED WORK

**Preference-based RL** Enabling development and comparison of PbRL algorithms, Kim et al. (2023) provided both expert-human labeled and script-labeled preference datasets for D4RL offline benchmark tasks. They also proposed the Preference Transformer (PT) reward model trained with a BT-based loss function to learn from preference data. As reward models can fail to capture true underlying preferences with limited data, subsequent works developed methods that avoid learning a reward model (Hejna & Sadigh, 2023; Hejna et al., 2024; An et al., 2023; Kang et al., 2023; Kuhar et al., 2023; Kim et al., 2024; Zhang et al., 2024). Inverse Preference Learning (Hejna & Sadigh, 2023) reformulates the BT model in terms of the RL Q-function, and can be used both in online and offline learning. In a similar vein of avoiding reward modeling, Offline Preference-guided Policy Optimization (OPPO) learns a trajectory encoder, an optimal latent, and learns a Decision Transformer policy conditioned on the latents (Kang et al., 2023).

**Contrastive learning in PbRL** Contrastive Preference Learning (CPL) generalizes the BT model and uses contrastive learning on the discounted sum of log policy for preferred and non-preferred segments (Hejna et al., 2024). CPL reformulates policy learning as a supervised learning objective rather than RL. Direct Preference-based Policy Optimization (DPPO) learns a BT-based preference predictor network, infers preferences for a full offline dataset, and lastly conducts contrastive learning to align policy predictions with the inferred preferred trajectories (An et al., 2023). Learning to Discern (L2D) conducts contrastive learning between trajectories of different labels (Kuhar et al., 2023). They then train a network with a BT-based loss, and its output is mapped to labels to filter low quality trajectories for downstream Imitation learning (IL).

**Robustness in PbRL** Though robustness techniques have been studied extensively in supervised learning contexts (Wang et al., 2021; Zhang et al., 2018; Han et al., 2018; Lukasik et al., 2020; Song et al., 2023), relatively little attention has been given in PbRL to the effect of labeling noise. Cheng et al. (2024) developed a PbRL method to filter out noisy preferences by defining a time dependent threshold for KL-divergence between predicted preference and the provided label. However, this framework involves querying human preferences iteratively online during policy training; it is not straightforward to adapt to our setting, in which the preference set is fixed and new queries cannot be sampled. Sun et al. (2025) examine preference learning in an LLM context, and they showed theoretically that BT formulations are not necessary. Instead of a BT loss that predicts the probability of preferring one response over another, they propose a simple classifier approach of predicting binary response preference. Compared against BT based approaches, they showed improved performance on LLM human value metrics for label error rates above 10%.

Our work diverges from these previous works as follows: As discussed in Section 1, the vast majority of PbRL works rely on BT assumptions. Our work prioritizes representation learning and avoids BT-modeling due to the potential to overfit (Azar et al., 2024), especially problematic with noisy comparison labels (Sun et al., 2025). Unlike the methods that do not learn an explicit reward function (Kang et al., 2023; Hejna et al., 2024; An et al., 2023; Kuhar et al., 2023), we use our representations to provide rewards which enables versatility to off-the-shelf offline and online RL algorithms (advantageous as discussed in Section 1). Also, if the problem setup has a known task reward, as occurring in a robotics setting, our method allows easy reward shaping by adding task rewards to preference inferred rewards. The classifier approach proposed by Sun et al. (2025) lays out a theoretical foundation for a non-BT approach. However, they focus on the LLM bandit setting whereas we focus on RL environments, with multi-step state/action trajectories. Whereas they focus on label classification, we focus on representation learning and infer rewards cheaply afterwards.

## 3  SIMILARITY AS REWARD ALIGNMENT (SARA)

We first review the PbRL setup. We then describe the SARA model, comprising a contrastive transformer encoder to learn a preferred set representation and a reward inference method. Appendix F provides details and hyperparameters.

**Preliminaries** In the RL paradigm, an agent at timestep $t$ and state $\mathbf{s}_t$ interacts with the environment by choosing an action $\mathbf{a}_t$. The action is chosen via its policy $\mathbf{a}_t = \pi(\mathbf{s}_t)$ which is a mapping from state to action. The environment provides reward $r(\mathbf{s}_t, \mathbf{a}_t)$ and transitions the agent to the next state $\mathbf{s}_{t+1}$. RL algorithms aim to learn a policy that maximizes the discounted cumulative reward, $R_t = \sum_{k=0}^{\infty} \gamma^k r(\mathbf{s}_{t+k}, \mathbf{a}_{t+k})$ with discount factor $\gamma$.

To address the reward engineering problem (Sutton & Barto, 2018; Dann et al., 2023), PbRL leverages human labeled preferences to learn policies that align with human intent (Wirth et al., 2017). Several previous approaches (Christiano et al., 2017; Kim et al., 2023; Hejna & Sadigh, 2023; Hejna et al., 2024; An et al., 2023; Kang et al., 2023) assume that human feedback is given in the form of preferences over trajectory pairs. Each trajectory segment $\sigma$ consists of $H$ state-action transitions: $\sigma = \{(s_0, a_0), (s_1, a_1), \ldots, (s_{H-1}, a_{H-1})\}$. Given a pair of segments $(\sigma^0, \sigma^1)$, a human annotator provides a preference label $y \in \{0, 0.5, 1\}$. The labels $y = 0$ and $y = 1$ indicate $\sigma^0 \succ \sigma^1$ and $\sigma^1 \succ \sigma^0$, respectively. The neutral preference $y = 0.5$ designates equal preference between the two trajectories.

**SARA contrastive encoder** The SARA encoder produces a single latent representation of all preferred labeled trajectories in the preference dataset. The SARA encoder addresses noisy preference learning by learning robust set-level representations that distinguish preferred from non-preferred behaviors, rather than relying on potentially unreliable pairwise comparisons.

We assume access to two sets of trajectories: a set of preferred trajectories and a set of non-preferred trajectories. In contrast to standard approaches, we do not require trajectories to be given in pairs, allowing the sets to have different sizes. When working with datasets that provide labeled pairs, we break apart the pairs to form these two sets, discarding the specific pairwise rankings. For pairs with neutral preference ($y = 0.5$), we include both trajectories in both sets.

Our contrastive encoder processes trajectories through a two-stage architecture. In the first stage, each trajectory passes through Transformer Encoder 1 (Figure 1) with positional encoding of time, followed by average pooling over timesteps to produce a single encoding per trajectory. This yields trajectory encodings $\mathbf{u}_{p,i}$ and $\mathbf{u}_{n,i}$ for preferred and non-preferred trajectories, respectively.

In the second stage, we randomly partition trajectory encodings within each category (preferred/non-preferred) into $k = 2$ subsets. Each subset then passes through Transformer Encoder 2, allowing trajectories within the same subset to attend to each other. This produces set-level encodings $\mathbf{z}_p^k$ and $\mathbf{z}_n^k$ for each preferred and non-preferred subset, respectively. We need a minimum of $k = 2$, so that we have at least one positive example for the contrastive loss. The model shows low sensitivity to the choice of $k$ provided sufficient trajectories exist in each subset (see Appendix A).

We train the encoder using the SimCLR contrastive loss (Chen et al., 2020) to pull together preferred subset representations $\mathbf{z}_p^k$ while pushing apart preferred and non-preferred representations. Let the outputs of the SARA encoder be $P = \{z_p^1, ..., z_p^k\}$ and $N = \{z_n^1, ..., z_n^k\}$ and cos denote cosine similarity.

Then SimCLR loss is given as:

$$\mathcal{L}_{\text{total}} = \frac{1}{2k} \left( \sum_{i=1}^{k} \mathcal{L}(z_p^i) + \sum_{i=1}^{k} \mathcal{L}(z_n^i) \right).$$

where for example,

$$\mathcal{L}(z_p^i) = -\frac{1}{k-1} \sum_{j \neq i} \log \frac{\exp\big(\cos(z_p^i, z_p^j)/\tau\big)}{\sum_{j \neq i} \exp\big(\cos(z_p^i, z_p^j)/\tau\big) + \sum_{\ell=1}^{k} \exp\big(\cos(z_p^i, z_n^\ell)/\tau\big)}.$$

The $\mathcal{L}(z_n^i)$ is defined analogously.

We randomize the composition of trajectories in each subset at every training epoch. This randomization strategy forces the encoder to learn generalizable patterns that distinguish preferred from non-preferred behavior rather than overfitting to specific trajectory pairings.

This approach provides several key advantages. The transformer architecture naturally develops robustness to mislabeled or ambiguous trajectories by learning to downweight patterns that do not reliably distinguish preference categories. Additionally, the architecture handles variable numbers of trajectories within each set. This allows us to train on subsets and then feed in the full set of preferred trajectories at inference.

**SARA reward inference** At inference time, we encode the complete set of preferred trajectories to obtain $\mathbf{z}_p^*$, our fixed preference representation. This frozen preferred representation then serves as the basis for computing similarity rewards in downstream tasks, providing a stable reference point that captures the essential characteristics of preferred behavior patterns. For each trajectory up to time $t$, we get the latent $\mathbf{z}_t = \mathcal{E}(\sigma_t)$, where $\mathcal{E}$ is the trained and frozen encoder. Then we simply compute the reward at time $t$ as: $r_t = \cos(\mathbf{z}_t, \mathbf{z}_p^*)$ using our frozen preferred latent (Figure 1, Step 3). This is a simple yet novel proposal for reward estimation from preferences.

**SARA reward: theoretical justification** Sun et al. (2025) examined preference learning in a LLM context, and they noted that BT formulations are not necessary. Instead of a BT loss that predicts the probability of preferring one response over another, they proposed a simple classifier approach of predicting preference (1 or 0 labeling) for the responses. They showed that such an approach preserves the ordering of the underlying true reward function, and that it is sufficient for downstream LLM alignment. Thus, instead of predicting the BT $P[\sigma^1 \succ \sigma^0; \psi]$ for a reward model parameterized by $\psi$, it is sufficient to predict the probability to be preferred $P[\sigma^i; \psi]$ with $i \in \{0, 1\}$ (Sun et al., 2025). In the next paragraph, we show that our approach implicitly does the same.

While our work focuses on representation learning followed by reward inference, their model focuses on classification for learning an explicit reward model. Nonetheless, their work provides theoretical grounding for our proposal that we learn based on individual trajectory labelings of preferred vs. non-preferred rather than learning the BT-based relative rankings. After training SARA, we conduct inference on a newly sampled trajectory $\sigma_t$. We pass $\sigma_t$ through our trained SARA encoder to get $z_t$. We then propose the following model to estimate the probability of the sampled trajectory $\sigma_t$ being preferred, given its latent representation:

$$ P\big(p \mid \mathbf{z}_t\big) = \frac{\exp\big(\cos(z_t, z_p^*)\big)}{\exp\big(\cos(z_t, z_p^*)\big) + \exp\big(\cos(z_t, z_n^*)\big)} = \frac{1}{1 + \exp\Big(-\big[\cos(z_t, z_p^*) - \cos(z_t, z_n^*)\big]\Big)}. $$

Such a probability function is a natural choice because the SimCLR loss aligns and separates latents using exponentiated cosine similarities. In RL we want to incentivize actions that have high probability of being preferred. Therefore, we simply set our reward equal to $r_t = \cos(\mathbf{z}_t, \mathbf{z}_p^*) - \alpha \cos(\mathbf{z}_t, \mathbf{z}_n^*)$, where $\alpha \geq 0$ is a hyperparameter to control the trade-off between the two terms. Empirically, we found $\alpha = 0$ to be optimal in all our experiments (both the offline and the online reward shaping experiments). With that, we recover our proposed reward in Section 3. Similar to Sun et al. (2025), we score trajectories on their alignment with preferred trajectories rather than relying on potentially noisy relative labels.

Our approach represents a departure from pairwise modeling of the BT model. We provide mechanistic justifications in Section 5.

## 4 OFFLINE RL EXPERIMENTS

In this section we address the following questions: First, how does using SARA inferred rewards compare to prior PbRL algorithms in the domain of offline RL? Secondly, how does SARA perform when the dataset is modified, *i.e.* neutral preferences are excluded or labeling mistakes occur?

**Setup** Similar to past works (Kim et al., 2023; An et al., 2023), we evaluate our framework in the offline setting on the following D4RL benchmark datasets: Mujoco locomotion (4 datasets), Franka Kitchen (2 datasets), Adroit (2 datasets) (Fu et al., 2021; Gupta; muj). For the Mujoco and

Table 1: Average normalized policy evaluation rewards (8 seeds) under different human preference mistake rates. Values in **bold** are the highest per row; underlined are within 1% of the best. The $\pm$ denotes standard deviation.

| Task | Err Rate | Oracle | PT | PT+ADT | DPPO | SARA |
|---|---|---|---|---|---|---|
| hopper-med-replay | 0% | 92.26 $\pm$13.6 | 74.48 $\pm$21.3 | 80.24 $\pm$16.1 | 68.98 $\pm$18.4 | **84.68** $\pm$**3.1** |
|  | 20% |  | 49.77 $\pm$25.7 | 62.87 $\pm$24.0 | 68.67 $\pm$19.8 | **82.94** $\pm$**5.8** |
| hopper-med-expert | 0% | 80.82 $\pm$44.5 | 89.64 $\pm$28.3 | 71.33 $\pm$40.6 | **108.09** $\pm$**10.8** | 80.45 $\pm$48.1 |
|  | 20% |  | 68.14 $\pm$36.2 | 79.02 $\pm$21.0 | 28.92 $\pm$29.1 | **85.16** $\pm$**17.0** |
| walker2d-med-replay | 0% | 77.53 $\pm$15.5 | 74.43 $\pm$8.0 | 75.68 $\pm$8.3 | 47.21 $\pm$28.0 | **78.21** $\pm$**5.8** |
|  | 20% |  | 71.73 $\pm$10.7 | 74.42 $\pm$12.6 | 41.00 $\pm$26.8 | **76.29** $\pm$**13.2** |
| walker2d-med-expert | 0% | 107.57 $\pm$8.5 | 109.74 $\pm$1.1 | **109.98** $\pm$**1.0** | 108.73 $\pm$0.4 | 108.35 $\pm$5.4 |
|  | 20% |  | 109.37 $\pm$1.5 | **109.53** $\pm$**1.6** | 108.78 $\pm$0.4 | 108.37 $\pm$5.7 |
| halfcheetah-med-replay | 0% | 42.68 $\pm$2.5 | 40.94 $\pm$2.7 | **42.85** $\pm$**1.7** | 39.94 $\pm$4.3 | 41.65 $\pm$2.0 |
|  | 20% |  | 41.16 $\pm$1.9 | **42.25** $\pm$**2.2** | 38.59 $\pm$6.9 | 42.08 $\pm$2.1 |
| halfcheetah-med-expert | 0% | 86.26 $\pm$14.1 | 86.62 $\pm$14.2 | 89.46 $\pm$9.4 | **92.18** $\pm$**8.5** | 86.56 $\pm$13.5 |
|  | 20% |  | 87.97 $\pm$11.1 | 89.18 $\pm$10.4 | **92.32** $\pm$**7.6** | 88.41 $\pm$10.4 |
| kitchen-partial | 0% | 44.88 $\pm$31.4 | 59.45 $\pm$15.6 | 61.68 $\pm$15.2 | 40.39 $\pm$18.9 | **64.84** $\pm$**13.2** |
|  | 20% |  | 58.55 $\pm$18.6 | 60.55 $\pm$16.7 | 38.44 $\pm$19.3 | **64.18** $\pm$**15.4** |
| kitchen-mixed | 0% | 54.02 $\pm$16.4 | 53.32 $\pm$9.8 | **53.48** $\pm$**9.7** | 43.63 $\pm$17.9 | 50.51 $\pm$6.4 |
|  | 20% |  | 44.65 $\pm$16.9 | 41.60 $\pm$20.5 | 46.05 $\pm$18.4 | **49.02** $\pm$**13.5** |

Adroit tasks, we use the preference datasets provided by Kim et al. (2023). We use the datasets by An et al. (2023) for the Kitchen tasks. All preference datasets comprise a limited subset of labeled trajectory pairs (100-500 pairs, depending on the dataset) relative to the full number of offline trajectories. The Adroit and Kitchen tasks have high dimensional state/action spaces (69 state+action dimensions) relative to the Mujoco tasks (14-23 state+action dimensions). Thus, our experiments comprise a variety of task environments, labeler sources, and state-action dimensionalities. We did not experiment on AntMaze due to a critical bug noted by An et al. (2023) (Appendix I). Additional details on the preference and full offline datasets can be found in Appendix E.1. The policy evaluation rewards exhibit high variance and are quite similar across models for the Adroit tasks, so we defer the Adroit results to Appendix B.3.

**Evaluation Metrics** After training on preference datasets, we infer the rewards $r_t$, for all transitions in the full offline dataset as discussed in Section 3. We then evaluate alignment of our learned rewards to the given preference criteria via two metrics: 1) Pearson correlation between inferred rewards and the environment rewards and 2) policy evaluation returns after offline RL training with the reward inferred offline dataset. The first metric, used also by Choi et al. (2024); Zhang et al. (2024); Liu et al. (2025), is valid because the preference criteria given to the human labelers aligns closely to the crafting of the environment reward functions. For example, the hopper environment rewards in proportion to the velocity of its movement and remaining in a stable upright position hop (b) Likewise, the preference criteria is given that hopper moves as far as possible and lands steadily (Appendix C of Kim et al. (2023)). The human preference criteria and the environment reward function incentivize the same behavior, so the correlation to the environment reward is a proxy for adherence to the preference labels.

For the policy evaluation returns, we conduct offline RL training with the state-of-the art Implicit Q-Learning (IQL) algorithm (Kostrikov et al., 2022), as in several prior PbRL works (Kim et al., 2023; Kostrikov et al., 2022). We adapt the OfflineRL-kit IQL implementation for our purposes (Sun, 2023), and we match preprocessing steps and hyperparameters to the recommended values in (Kostrikov et al., 2022; Kim et al., 2023) (Appendix F). We train SARA+IQL and baselines on 8 seeds with multiple evaluation episodes (see Appendix G.2 for reward normalization method and reward reporting method). We also provide our results for the oracle IQL, which uses the true environmental rewards rather than preference based rewards.

We compare against the following baselines. The first two baselines (PT, PT+ADT) learn a reward model from the preference dataset and then conduct IQL training. The last baseline does not learn an explicit reward model (DPPO).

- **Preference Transformer (PT):** As in our model, PT uses a transformer backbone (Kim et al., 2023). Unlike our model, PT learns an explicit reward model with a BT based loss.

- **PT with Adaptive Denoising Training (PT+ADT):** We introduce this novel application of ADT as a baseline. In each training step, ADT drops a $\tau(t)$ fraction of queries with the largest cross-entropy loss, where $\tau(t) = \min(\gamma t, \tau_{\max})$ (Wang et al., 2021). We set $\tau_{\max} = 0.3$ and $\gamma = 0.003$ for our datasets. Prior works have considered ADT in the setting of iterative online human feedback (Cheng et al., 2024), but to our knowledge we are the first to apply ADT to learning the PT reward model.

- **Direct Preference-based Policy Optimization (DPPO):** As in our model, DPPO relies on contrastive learning and does not learn an explicit reward model. However, the contrastive learning is used to learn the policy directly and aims to align policy predictions with inferred preferred trajectories (An et al., 2023) (see Section 2 for additional details).

**Preference data with labeling noise** On the original unmodified preference sets, SARA either outperforms or is on-par with baseline methods (Table 1). We take these preference datasets and randomly flip 10%, 20%, and 40% of the non-neutral labels. These error rates are in accordance with realistic error rates noted in prior literature, as discussed in Section 1 (page 2 top). As shown in Table 1, the IQL policy with SARA computed rewards substantially outperforms baselines in the 20% case. Due to space constraints, we show results for the 10% and 40% error rates in Appendix B.2.

Table 7 shows variation in model performance as a result of tuning error rates between 0-40%. Our method's robustness is evidenced by the consistency it shows as noise rate is varied. For example, PT's performance drops from 74.48 to 49.77 as mistake rate goes from 0 to 20%. On the other hand, SARA only drops from 84.66 to 82.94. Likewise, DPPO has impressive performance on the hop-medium-expert datasets at low (0, 10%) error rates). We infer that DPPO is able to match expert trajectories in such datasets with many examples. However, DPPO suffers tremendously by dropping to 28.92 at 20% error rate. Finally, our novel implementation of PT+ADT also provides significant improvement over PT.

Table 2: Mean normalized policy evaluation rewards across six preference labeling conditions (4 error rates, excluding neutral preferences, and script labeled). The $\pm$ denotes standard deviation across labeling conditions, and it accounts for both seed variance and labeling variance. We bold the highest mean and lowest standard deviation in each row.

| Task | PT | PT+ADT | DPPO | SARA |
|------|-----|--------|------|------|
| hopper-med-replay | $65.73 \pm 25.49$ | $71.29 \pm 24.02$ | $57.31 \pm 26.71$ | $\mathbf{81.49 \pm 14.64}$ |
| hopper-med-expert | $68.73 \pm 36.17$ | $77.09 \pm \mathbf{33.96}$ | $83.92 \pm 36.05$ | $\mathbf{85.22} \pm 34.68$ |
| walker2d-med-replay | $72.35 \pm 12.65$ | $72.58 \pm 16.25$ | $43.93 \pm 28.13$ | $\mathbf{75.95 \pm 11.29}$ |
| walker2d-med-expert | $107.10 \pm 9.41$ | $106.42 \pm 10.82$ | $\mathbf{108.73 \pm 0.42}$ | $108.70 \pm 4.57$ |
| kitchen-partial | $59.69 \pm 17.90$ | $60.76 \pm \mathbf{17.06}$ | $38.50 \pm 18.51$ | $\mathbf{61.01} \pm 17.93$ |
| kitchen-mixed | $\mathbf{49.85} \pm 15.53$ | $48.57 \pm 16.21$ | $45.68 \pm 17.97$ | $49.03 \pm \mathbf{11.40}$ |

**Robustness to dataset variants** Here we analyze our model's consistency across varying preference labeling conditions as further evidence of robustness. In addition to the four error rates, we also examined two additional labeling conditions: excluding neutral queries and labeling by a script oracle. In many realistic applications, labelers are often presented with queries where the two options are quite similar. In some designs, for example current OpenAI GPT models, the labeler is forced to pick a preferred option. In others, as in the dataset by Kim et al. (2023), the labelers are allowed to indicate indifference (Appendix E.1 provides percentage of neutral queries per dataset). The performance of the learned policy should ideally not have strong sensitivity to such choices. A PbRL framework exhibits robustness by its ability to discern patterns of preferences, and those learned patterns should not be contingent on whether or not neutrality is allowed. Script labeling, in which an oracle with knowledge of task rewards picks binary preferences based on comparisons of total returns, has also been examined in prior works (Kim et al., 2023; Zhang et al., 2024; Kang et al., 2023; Christiano et al., 2017; Cheng et al., 2024). As human labelers are known to disagree at rates up to 25% (Dubois

Table 3: Pearson correlation coefficients between transition rewards and environment provided rewards (unknown at training). Values are averages ± standard deviation are shown.

| Task | Error Rate | PT | PT+ADT | SARA |
|---|---|---|---|---|
| hopper-med-replay | 0% | 0.04 (±0.03) | 0.08 (±0.06) | **0.39** (±**0.04**) |
| | 20% | 0.03 (±0.03) | 0.04 (±0.05) | **0.36** (±**0.05**) |
| hopper-med-expert | 0% | -0.09 (±0.09) | 0.14 (±0.10) | **0.55** (±**0.14**) |
| | 20% | 0.08 (±0.07) | 0.02 (±0.13) | **0.14** (±**0.55**) |
| walker2d-med-replay | 0% | 0.34 (±0.08) | 0.50 (±0.06) | **0.73** (±**0.02**) |
| | 20% | 0.19 (±0.03) | 0.36 (±0.07) | **0.56** (±**0.20**) |
| walker2d-med-expert | 0% | 0.50 (±0.09) | 0.73 (±0.01) | **0.84** (±**0.05**) |
| | 20% | 0.11 (±0.07) | 0.48 (±0.10) | **0.81** (±**0.05**) |
| kitchen-partial | 0% | 0.22 (±0.09) | 0.13 (±0.07) | **0.39** (±**0.36**) |
| | 20% | -0.10 (±0.06) | -0.15 (±0.11) | **0.30** (±**0.28**) |
| kitchen-mixed | 0% | 0.01 (±0.08) | -0.10 (±0.11) | **0.06** (±**0.35**) |
| | 20% | -0.26 (±0.17) | -0.44 (±0.06) | **0.03** (±**0.35**) |

et al., 2023; Coste et al., 2024), performance on script labeling provides another method of comparing models. See Appendix B.2 for tables with exact values and error bars.

To measure robustness across labeling conditions, we show the mean normalized episode evaluation returns and standard deviation across the six labeling conditions (Table 2). We bold the *highest* mean and *lowest* standard deviation in each row. While baselines may exhibit better performance in mean or standard deviation as one-offs, SARA is the method that provides robustness most frequently. In cases where SARA loses compared to baselines, the margin by which it loses is quite small. We believe this evidences SARA's consistency in response to perturbations of the preference dataset.

**Correlation with environment transition rewards** SARA demonstrates substantially better Pearson correlation with environment rewards compared to baselines at all error rates (Table 3, 6). We conducted this analysis on full offline datasets at the transition level. Note that DPPO is excluded as it is a reward-free method.

As detailed in the Evaluation paragraph of this section, this metric provides insight on alignment to the given preference labeling criteria. However, this correlation advantage does not always result in a better policy. SARA's better correlation consistently translates to better policy performance on hopper replay and walker replay datasets, where it achieves higher policy rewards with far lower variance than PT and PT+ADT. However, on the hopper-medium-expert at 0%, PT correlates poorly to the environmental transition level rewards but does better than the IQL oracle in evaluation policy returns. We think that, due to the high annotation quality on this dataset with many expert trajectories, PT provides a reward model that leads to better policy learning in the IQL algorithm than the environment provided rewards. This comparison highlights that PbRL reward models on high quality expert data can drive better policy learning than the environment rewards.

On the other hand, SARA still learns distinguishing patterns of preference in the hopper expert case. By focusing on consensus signals, SARA likely sacrifices the ability to learn the better preference reward model for policy learning that PT was able to learn in this case of high annotation quality. However, SARA benefits by focusing on these general patterns in that its policy performance does not degrade as annotation quality decreases. Thus, SARA does not displace the benefits of BT reward models in such high quality data scenarios, but rather it offers an alternative and more robust learning mechanism in the often occurring case of non-expert data.

## 5 A MECHANISTIC ANALYSIS OF SARA'S ROBUSTNESS TO NOISE

A theoretical proof for both the BT and SARA models in noisy settings would require a statement about the convergence of optimal solutions on data with label noise, which is a non-convex optimization problem. Currently, formal convergence proofs for preference learning under noise represents

a significant open challenge in the field. The difficulty of providing a such a proof is evidenced by BT model literature itself. The BT model was originally proposed for ranking sports teams, and its theoretical foundation in PbRL is underexplored (Sun et al., 2025), despite the prominent applications in LLM fine tuning. Likewise, we cannot supply a proof either, but we provide reasoning for SARA's robustness compared to BT models, followed by empirical validation with ablation experiments.

The BT model learns an underlying reward model that captures preference for one **individual** trajectory over another. The key difference is that SARA intentionally only focuses on what makes the **set** of preferred different from the set of non-preferred, and it does not prioritize learning a good representation at the individual trajectory level. The focus then is not about learning to represent an individual trajectory but rather **learning which patterns to ignore and which to attend**.

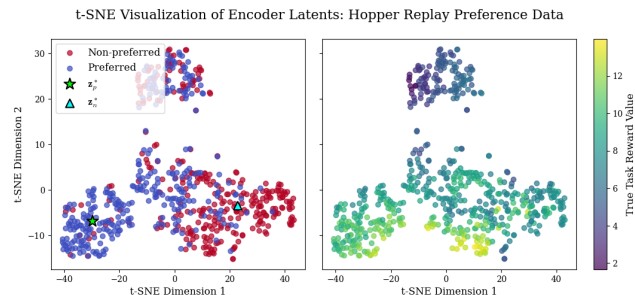

Figure 2: T-SNE embedding of the latents for the hopper-medium-replay-v2 preference data, either colored by preference (left) or true reward (right).

In the BT approach, a reward model is learned explicitly by maximizing likelihood of observed preferences: $\max_r \sum_{(i,j)} y_{ij} \log P[\sigma_i \succ \sigma_j; r]$ . Then each mislabeled pair $(i, j)$ directly corrupts the gradient: $\nabla_r \log P[\sigma_j \succ \sigma_i; r] = -\gamma \nabla_r \log P[\sigma_i \succ \sigma_j; r]$. The coefficient $\gamma = \frac{\text{sigmoid}(r(\sigma_i) - r(\sigma_j))}{\text{sigmoid}(-(r(\sigma_i) - r(\sigma_j)))}$ is always positive and blows up for large reward differences.

By contrast, SARA does not model individual comparisons. Its **set-based** contrastive objective encourages the encoder to learn **aggregate patterns** that consistently distinguish the preferred set from the non-preferred set. Even if some trajectories in the preferred set are mislabeled, the encoder must still produce a set representation $z_p$ that is dissimilar from the non-preferred set. To do so, the contrastive objective drives the following:

- **Encoder 1** places low attention weight on portions of mislabeled trajectories that resemble the non-preferred set. It focuses instead on transitions shared by the majority of correctly labeled preferred trajectories.
- **Encoder 2** aggregates these per-trajectory latents into a consensus representation that maximally separates the two sets.

In this way, the contrastive objective naturally aggregates to a dominant structure in the preferred set, rather than directly following gradients of individual noisy labels as BT does. Even when mislabeling occurs, the resulting $z_p^*$ is still a latent that shows low similarity to the non-preferred sets and high similarity to subsets of preferred.

This is underscored by inspecting visual embeddings of the learned representations. Figure 2 shows t-SNE embeddings of SARA learned latents for the human labeled hopper-medium-replay-v2 preference dataset from Kim et al. (2023). As encouraged by the contrastive learning objective, the encoder achieves good separation and clustering between *most* preferred and non-preferred trajectories, and the group embeddings $\mathbf{z}_p^*$ and $\mathbf{z}_n^*$ are centrally located within their respective clusters. However, some trajectories are not close to $\mathbf{z}_p^*$ nor $\mathbf{z}_n^*$, though they are labeled as such. Comparing the plots in Figure 2, we see that these separated trajectories exhibit a low environmental trajectory reward (unknown to SARA). Therefore, the SARA encoder can learn overall patterns but it does not artificially align poor quality trajectories to the preferred set even when the human labelers designate them as preferred.

**Ablation** We claim our set based encoding with contrastive learning is the key behind our robustness. To support this claim, we ablate the set idea and do contrastive learning on the individual trajectory representations. Our ablation, BT Contrastive, results in significantly lower policy returns at varying label error rates (Table 4). BT Contrastive includes the following steps: 1) The transformer encodes the trajectories $\boldsymbol{\sigma}_{p,i}$ and $\boldsymbol{\sigma}_{n,i}$ to $\mathbf{u}_{p,i}$ and $\mathbf{u}_{n,i}$. The contrastive learning is done between all pairs of individual trajectory encodings (not set encodings). Then 2) we learn a BT reward model, $r_\psi$, using

Table 4: Comparison of SARA and BT Contrastive across datasets with different error rates.

| Task | Error Rate | SARA | BT Contrastive |
|---|---|---|---|
| hopper-med-replay | 10% | **83.66** (±**3.5**) | 65.21 (±22.3) |
| | 20% | **82.94** (±**5.8**) | 64.19 (±22.0) |
| | 40% | **65.82** (±**28.9**) | 60.03 (±25.8) |
| hopper-med-expert | 10% | **84.95** (±**32.4**) | 80.65 (±29.4) |
| | 20% | 85.16 (±17.0) | **86.93** (±**21.8**) |
| | 40% | **83.38** (±**26.2**) | 79.92 (±27.9) |
| walker2d-med-replay | 10% | **78.18** (±**7.9**) | 66.13 (±15.2) |
| | 20% | **76.29** (±**13.2**) | 57.17 (±23.0) |
| | 40% | **68.32** (±**18.6**) | 64.94 (±16.1) |
| walker2d-med-expert | 10% | 108.66 (±3.7) | **109.51** (±**0.7**) |
| | 20% | 108.37 (±5.7) | **109.34** (±**1.3**) |
| | 40% | **109.02** (±**4.2**) | 107.90 (±9.2) |

the given preference set labels and their learned latents: $P[\boldsymbol{\sigma}_{p,i} \succ \boldsymbol{\sigma}_{n,i}; \psi] = P[\mathbf{u}_{p,i} \succ \mathbf{u}_{n,i}; \psi]$. In step 1 we match the encoder capacity to the original SARA encoder.

As an exception to the performance degradation, we see that BT Contrastive is on par with SARA on hopper-medium-expert at 20%. This result consistent with analysis by Sun et al. (2025), who notes that BT models can still perform well when annotation quality is high.

**Data Scaling** We reduce the amount of preference data by half. Appendix D shows that in the low data regime (50% of preference data) with label noise (20% error rate), SARA provides a substantial robustness advantage against the Bradley-Terry based PT model. However, our experiment shows SARA is less performant against PT in the low data regime with 0% error.

## 6 ADDITIONAL EXPERIMENTAL APPLICATIONS

We leverage our framework for additional application areas: cross-task preference transfer and online RL with reward shaping. Due to space constraints, we provide experiment details and results in Appendix C.

## 7 CONCLUSION

**Summary**    SARA is a novel algorithm that prioritizes robustness by estimating preference-based rewards via similarity with a contrastively learned latent. Rather than relying on BT-based rewards, SARA assumes presence of noisy labeling and learns a representation of preferences. SARA shows comparable or improved performance over SOTA baselines on preference sets between 0-40% label error rate and consistent performance across these variants. SARA outperforms BT based models on correlation with environmental rewards. We leverage SARA's strong reward estimates for additional applications, such as online RL and differing feedback format (Appendix C).

**Limitations and future work**    Our problem setup is one specific preference criteria per task; Kim et al. (2023); An et al. (2023) provide the criteria given to their labelers. However, SARA can be straight-forwardly applied towards the problem of heterogenous preference criteria. For our problem setup, we assumed two subset categories for Transformer Encoder 2, *preferred* and *non-preferred*. With multiple criteria, such as *preferred-fast*, *preferred-slow*, *non-preferred*, we would now have three categories which we train contrastively. Testing this would require creating multi-criteria RL preference datasets; we recommend this as a direction for future work.

To apply SARA to LLMs, we suggest adapting SARA for querying human preferences iteratively online during policy training. Doing so would require fine-tuning the encoder online as a human labeler provides feedback throughout policy training.

## 8 REPRODUCIBILITY STATEMENT

In order to ensure reproducibility we have taken the following steps. Sections F and G.1 provide details of architecture and hyperparameters. Upon acceptance, we plan to link our github repository in the paper. In addition to providing source code for our own model, our github repository additionally provides a user friendly pipeline script to train our model, the baselines, and the oracle on all our seeds and on all our dataset variants. By doing so we facilitate reproducibility of our own work as well as baseline models.

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

## A SENSITIVITY TO NUMBER OF SUBSETS PER CATEGORY (PARAMETER K)

As detailed in Section 5, our key innovation is to learn a representation for the **set** of preferred trajectories. The Transformer Encoder 2 of Figure 1 encodes a set of trajectories, where now trajectory representations within the set can attend to one another. We need at least one positive set encoding for each category (preferred and non-preferred) for the contrastive loss. This necessitates that we divide up the preferred trajectory encodings and non-preferred trajectory encodings in at minimum k=2 subsets. In each epoch, we shuffle the compositions of trajectories in each subset to avoid overfitting the representation to an exact subset composition. We used k=2 in all our experiments, but here we ablate the choice of k and show the model is not sensitive when k is low. We note that the k value is used as a training hyperparameter, not as an evaluation parameter. After training, we set k=1 (one set of preferred) and feed all preferred trajectories through the SARA encoder to infer $\mathbf{z}_\mathbf{p}^*$.

Table 5: Average normalized policy evaluation rewards (across 8 seeds), using human-labeled preference data (without mistakes). We vary the number of subsets $k$ per category during training the SARA encoder. The $\pm$ denotes standard deviation.

| Task | $k = 2$ | $k = 3$ | $k = 4$ | $k = 16$ |
|---|---|---|---|---|
| hopper-medium-replay-v2 | 84.68 $(\pm 3.1)$ | 85.00 $(\pm 3.3)$ | 84.48 $(\pm 4.0)$ | 76.78 $(\pm 18.9)$ |

When k is equal to the number of trajectories in each category (preferred or non-preferred), then one set is just a single trajectory. This is equivalent to contrastive learning on individual trajectories rather than sets of trajectories, as done in our ablation experiment in Section 5.

As we approach large k, we are approaching the regime of of contrastive learning between individual trajectories. Thus we see significant performance degradation at k=16. The results are not sensitive for lower values of k (i.e. 2,3,4).

## B ADDITIONAL OFFLINE RL RESULTS

### B.1 PEARSON CORRELATIONS WITH ENVIRONMENT REWARDS

The main paper provides results at 0% and 20% error rates. Here we provide the additional Pearson correlations for models trained on preference data with 10% and 40% error rates as well.

### B.2 LOCOMOTION AND KITCHEN TASKS

The main paper provides the tables for human-labeled preference sets with 0 and 20% error. Here we provide the tables for human-labeled preference sets with 10 and 40% error. We also show the results without neutral preferences and script labeled preference sets.

### B.3 OFFLINE RL RESULTS ON ADROIT TASKS

We also applied our framework to the Adroit tasks, with preference datasets for pen-cloned-v1 and pen-human-v1 provided by Kim et al. (2023). DPPO underperforms compared to other methods on pen-human-v1, but otherwise the mean evaluation policy rewards are similar across models (Tables 10, 11, 12, and 13). Variance is also quite high for all models. This is explained by within seed variance among the 10 evaluation episodes at each epoch, presumably due to randomized initial start states, as opposed to across seed variance. Among the 10 evaluation episodes at each epoch, we acquire maximum normalized episode returns of 179 and minimum returns between -2 to -4.

Table 6: Pearson correlation coefficients between transition rewards and environment provided rewards (unknown at training). Each model was trained across 8 seeds; averages $\pm$ standard deviation are shown, with highest values in each row in bold.

| Task | Error Rate | PT | PT+ADT | SARA |
|---|---|---|---|---|
| hop-medium-replay | 0% | 0.04 $(\pm 0.03)$ | 0.08 $(\pm 0.06)$ | **0.39** $(\pm 0.04)$ |
| | 10% | 0.08 $(\pm 0.03)$ | 0.06 $(\pm 0.08)$ | **0.19** $(\pm 0.08)$ |
| | 20% | 0.03 $(\pm 0.03)$ | 0.04 $(\pm 0.05)$ | **0.36** $(\pm 0.05)$ |
| | 40% | -0.04 $(\pm 0.02)$ | -0.03 $(\pm 0.06)$ | **0.02** $(\pm 0.22)$ |
| hop-medium-expert | 0% | -0.09 $(\pm 0.09)$ | 0.14 $(\pm 0.10)$ | **0.55** $(\pm 0.14)$ |
| | 10% | -0.01 $(\pm 0.08)$ | 0.16 $(\pm 0.08)$ | **0.28** $(\pm 0.36)$ |
| | 20% | 0.08 $(\pm 0.07)$ | 0.02 $(\pm 0.13)$ | **0.14** $(\pm 0.55)$ |
| | 40% | 0.04 $(\pm 0.05)$ | 0.06 $(\pm 0.14)$ | **0.13** $(\pm 0.38)$ |
| walk-medium-replay | 0% | 0.34 $(\pm 0.08)$ | 0.50 $(\pm 0.06)$ | **0.73** $(\pm 0.02)$ |
| | 10% | 0.27 $(\pm 0.06)$ | 0.36 $(\pm 0.06)$ | **0.70** $(\pm 0.03)$ |
| | 20% | 0.19 $(\pm 0.03)$ | 0.36 $(\pm 0.07)$ | **0.56** $(\pm 0.20)$ |
| | 40% | 0.04 $(\pm 0.02)$ | 0.12 $(\pm 0.08)$ | **0.32** $(\pm 0.23)$ |
| walk-medium-expert | 0% | 0.50 $(\pm 0.09)$ | 0.73 $(\pm 0.01)$ | **0.84** $(\pm 0.05)$ |
| | 10% | 0.33 $(\pm 0.07)$ | 0.65 $(\pm 0.05)$ | **0.86** $(\pm 0.03)$ |
| | 20% | 0.11 $(\pm 0.07)$ | 0.48 $(\pm 0.10)$ | **0.81** $(\pm 0.05)$ |
| | 40% | -0.14 $(\pm 0.03)$ | -0.09 $(\pm 0.05)$ | **0.51** $(\pm 0.37)$ |
| kitchen-partial | 0% | 0.22 $(\pm 0.09)$ | 0.13 $(\pm 0.07)$ | **0.39** $(\pm 0.36)$ |
| | 10% | 0.07 $(\pm 0.08)$ | -0.01 $(\pm 0.07)$ | **0.36** $(\pm 0.31)$ |
| | 20% | -0.10 $(\pm 0.06)$ | -0.15 $(\pm 0.11)$ | **0.30** $(\pm 0.28)$ |
| | 40% | -0.15 $(\pm 0.09)$ | -0.18 $(\pm 0.13)$ | **0.22** $(\pm 0.38)$ |
| kitchen-mixed | 0% | 0.01 $(\pm 0.08)$ | -0.10 $(\pm 0.11)$ | **0.06** $(\pm 0.35)$ |
| | 10% | -0.26 $(\pm 0.08)$ | -0.34 $(\pm 0.06)$ | **-0.14** $(\pm 0.33)$ |
| | 20% | -0.26 $(\pm 0.17)$ | -0.44 $(\pm 0.06)$ | **0.03** $(\pm 0.35)$ |
| | 40% | -0.24 $(\pm 0.08)$ | -0.49 $(\pm 0.07)$ | **0.02** $(\pm 0.33)$ |

## C ADDITIONAL APPLICATIONS TO SUPPORT VERSATILITY STATEMENTS

**Filtering low quality trajectories for downstream imitation learning** As ground truth rewards may be unknown, we examine whether the SARA encoder can identify low quality trajectories from preference data. Human preference labels and ground truth rewards frequently do not align (Kim et al., 2023), so many poor quality trajectories may be labeled preferred ($y = 1$). Figure 2 shows a t-SNE plot of SARA learned latents for the human labeled hopper replay preference dataset from Kim et al. (2023). We exclude the neutral queries from the original preference dataset, but we otherwise do not corrupt or modify the dataset in any way. As encouraged by the constrastive learning objective, the encoder achieves good separation and clustering between *most* preferred and non-preferred trajectories, and the group embeddings $\mathbf{z}_p^*$ and $\mathbf{z}_n^*$ are centrally located within their respective clusters. However, some trajectories are not close to $\mathbf{z}_p^*$ nor $\mathbf{z}_n^*$, though they are labeled as such. Comparing the plots in Figure 2, we see that these separated trajectories exhibit a low ground truth reward. Therefore, the SARA encoder can learn overall patterns but it does not artificially align poor quality trajectories to the preferred set even when the human labelers designate them as preferred. We can further exploit the encoder results to filter low quality trajectories. After training the encoder, we merely need to filter out trajectories with large distance in latent space from both $\mathbf{z}_p^*$ and $\mathbf{z}_n^*$. Kuhar et al. (2023) notes that filtering should lead to a better policy in downstream IL. As IL is outside the scope of this work, we defer such analysis to future works.

**Transfer of preferences to morphologically harder task** We investigate whether the preference dataset for a morphologically simple task can be used to infer rewards for a harder task. We take the preference dataset from Kim et al. (2023) for trajectories from hopper-medium-replay-v2. Our goal is to learn the SARA encoder on this preference set and then infer rewards for the walker-medium-

Table 7: Average normalized policy evaluation rewards (8 seeds) under different human preference mistake rates. Oracle IQL results are constant across mistake rates and shown once in grey. Values in **bold** are the highest per row; underlined are within 1% of the best. The $\pm$ denotes standard deviation.

| Task | Err Rate | Oracle | PT | PT+ADT | DPPO | SARA |
|---|---|---|---|---|---|---|
| hop-med-replay | 0% | 92.26 $\pm$13.6 | 74.48 $\pm$21.3 | 80.24 $\pm$16.1 | 68.98 $\pm$18.4 | **84.68** $\pm$**3.1** |
| | 10% | | **86.31** $\pm$**7.9** | 77.14 $\pm$18.5 | 70.76 $\pm$16.6 | 83.66 $\pm$3.5 |
| | 20% | | 49.77 $\pm$25.7 | 62.87 $\pm$24.0 | 68.67 $\pm$19.8 | **82.94** $\pm$**5.8** |
| | 40% | | 58.72 $\pm$17.9 | 53.87 $\pm$20.9 | 32.51 $\pm$23.2 | **65.82** $\pm$**28.9** |
| hop-med-expert | 0% | 80.82 $\pm$44.5 | 89.64 $\pm$28.3 | 71.33 $\pm$40.6 | **108.09** $\pm$**10.8** | 80.45 $\pm$48.1 |
| | 10% | | 78.54 $\pm$30.1 | 80.69 $\pm$27.3 | **100.32** $\pm$**20.4** | 84.95 $\pm$32.4 |
| | 20% | | 68.14 $\pm$36.2 | 79.02 $\pm$21.0 | 28.92 $\pm$29.1 | **85.16** $\pm$**17.0** |
| | 40% | | 53.10 $\pm$24.9 | 70.00 $\pm$29.2 | 55.11 $\pm$8.2 | **83.38** $\pm$**26.2** |
| walker2d-med-replay | 0% | 77.53 $\pm$15.5 | 74.43 $\pm$8.0 | 75.68 $\pm$8.3 | 47.21 $\pm$28.0 | **78.21** $\pm$**5.8** |
| | 10% | | 74.59 $\pm$6.7 | 64.42 $\pm$28.2 | 47.18 $\pm$27.0 | **78.18** $\pm$**7.9** |
| | 20% | | 71.73 $\pm$10.7 | 74.42 $\pm$12.6 | 41.00 $\pm$26.8 | **76.29** $\pm$**13.2** |
| | 40% | | 61.52 $\pm$19.3 | 67.86 $\pm$16.3 | 28.07 $\pm$27.9 | **68.32** $\pm$**18.6** |
| walker2d-med-expert | 0% | 107.57 $\pm$8.5 | 109.74 $\pm$1.1 | **109.98** $\pm$**1.0** | 108.73 $\pm$0.4 | 108.35 $\pm$5.4 |
| | 10% | | **109.89** $\pm$**1.1** | 109.85 $\pm$3.5 | 108.75 $\pm$0.4 | 108.66 $\pm$3.6 |
| | 20% | | 109.37 $\pm$1.5 | **109.53** $\pm$**1.6** | 108.78 $\pm$0.4 | 108.37 $\pm$5.7 |
| | 40% | | 93.69 $\pm$16.9 | 89.83 $\pm$18.7 | 108.55 $\pm$0.5 | **109.02** $\pm$**4.2** |
| kitchen-partial | 0% | 44.88 $\pm$31.4 | 59.45 $\pm$15.6 | 61.68 $\pm$15.2 | 40.39 $\pm$18.9 | **64.84** $\pm$**13.2** |
| | 10% | | 59.30 $\pm$18.2 | 60.27 $\pm$15.0 | 37.27 $\pm$17.5 | **62.50** $\pm$**14.2** |
| | 20% | | 58.55 $\pm$18.6 | 60.55 $\pm$16.7 | 38.44 $\pm$19.3 | **64.18** $\pm$**15.4** |
| | 40% | | 50.86 $\pm$21.7 | **58.67** $\pm$**18.7** | 36.84 $\pm$17.5 | 54.69 $\pm$23.8 |
| kitchen-mixed | 0% | 54.02 $\pm$16.4 | 53.32 $\pm$9.8 | **53.48** $\pm$**9.7** | 43.63 $\pm$17.9 | 50.51 $\pm$6.4 |
| | 10% | | **53.36** $\pm$**9.7** | 48.71 $\pm$13.8 | 44.96 $\pm$20.1 | 50.27 $\pm$7.9 |
| | 20% | | 44.65 $\pm$16.9 | 41.60 $\pm$20.5 | 46.05 $\pm$18.4 | **49.02** $\pm$**13.5** |
| | 40% | | **49.53** $\pm$**19.8** | 46.41 $\pm$20.5 | 49.30 $\pm$14.8 | 42.73 $\pm$16.3 |

Table 8: Average normalized policy evaluation rewards (across 8 seeds), using **human-labeled preference data without neutral preferences**. Values in **bold** are best (highest reward) in each row and underlined are within 1% of best. The $\pm$ denotes standard deviation.

| Task | IQL (Oracle) | PT | PT+ADT | DPPO | SARA |
|---|---|---|---|---|---|
| hopper-med-replay | 92.26 $\pm$13.6 | **86.67** $\pm$**4.7** | 83.06 $\pm$8.8 | 70.18 $\pm$19.9 | 84.43 $\pm$4.3 |
| hopper-med-expert | 80.82 $\pm$44.5 | 59.90 $\pm$46.2 | 74.53 $\pm$41.7 | **108.88** $\pm$**9.5** | 80.93 $\pm$43.9 |
| walker2d-replay | 77.53 $\pm$15.5 | 75.14 $\pm$3.9 | 76.69 $\pm$6.5 | 47.87 $\pm$27.6 | **78.21** $\pm$**5.8** |
| walker2d-med-expert | 107.57 $\pm$8.5 | **110.09** $\pm$**4.6** | 109.61 $\pm$2.2 | 108.77 $\pm$0.4 | 108.91 $\pm$3.4 |
| kitchen-partial | 44.88 $\pm$31.4 | 61.64 $\pm$15.0 | 60.78 $\pm$15.1 | 39.77 $\pm$18.9 | **63.79** $\pm$**14.6** |
| kitchen-mixed | 54.02 $\pm$16.4 | 50.66 $\pm$13.8 | **52.11** $\pm$**12.5** | 45.35 $\pm$17.9 | 51.80 $\pm$7.2 |

replay-v2 dataset. SARA reward inference relies on feeding walker trajectories into the learned encoder, so we map the state and action space dimensions of the hopper to that of the walker. We do so by crudely assuming that preferences for the hopper state-action trajectories can transfer to the additional degrees of freedom in the walker due to the symmetry of the joints (see Appendix H for details).

After doubling the joint angles, velocities, and torques in the hopper replay preference set, we then train the SARA encoder with this modified set. Next we take the full offline walker replay set and

Table 9: Average normalized policy evaluation rewards (across 8 seeds), using **script-labeled preference data**. Values in **bold** are best (highest reward) in each row and underlined are within 1% of best. The $\pm$ denotes standard deviation.

| Task | IQL (Oracle) | PT | PT+ADT | DPPO | SARA |
|---|---|---|---|---|---|
| hopper-med-replay | 92.26 $\pm$13.6 | 38.43 $\pm$19.8 | 70.55 $\pm$33.8 | 32.75 $\pm$22.6 | **87.43** $\pm$**8.4** |
| hopper-expert | 80.82 $\pm$44.5 | 63.03 $\pm$34.5 | 86.96 $\pm$35.9 | **102.19** $\pm$**24.4** | 96.43 $\pm$27.9 |
| walker2d-med-replay | 77.53 $\pm$15.5 | **76.69** $\pm$**14.0** | 76.40 $\pm$11.0 | 52.27 $\pm$24.7 | 76.51 $\pm$6.4 |
| walker2d-med-expert | 107.57 $\pm$8.5 | **109.80** $\pm$**1.9** | 109.74 $\pm$1.0 | 108.81 $\pm$0.4 | 108.88 $\pm$4.6 |
| kitchen-partial | 44.88 $\pm$31.4 | **68.32** $\pm$**12.2** | 62.58 $\pm$20.6 | 38.28 $\pm$18.6 | 56.05 $\pm$20.9 |
| kitchen-mixed | 54.02 $\pm$16.4 | 47.58 $\pm$18.2 | 49.10 $\pm$14.3 | 44.77 $\pm$17.8 | **49.84** $\pm$**11.2** |

Table 10: Average normalized policy evaluation rewards (across 8 seeds), using **human-labeled preference data with 20% label error rate**. Values in **bold** are best (highest reward) in each row. The $\pm$ denotes standard deviation.

| Task | IQL (Oracle) | PT | PT+ADT | DPPO | SARA |
|---|---|---|---|---|---|
| pen-human-v1 | 85.19 ($\pm$62.1) | 81.91 ($\pm$64.5) | 79.42 ($\pm$63.2) | 71.99 ($\pm$62.5) | **83.04** ($\pm$**63.6**) |
| pen-cloned-v1 | 81.49 ($\pm$62.6) | 69.30 ($\pm$63.8) | 67.66 ($\pm$64.1) | **72.64** ($\pm$**64.4**) | 69.78 ($\pm$63.0) |

Table 11: Average normalized policy evaluation rewards (across 8 seeds), using **human-labeled preference data with neutral preferences**. Values in **bold** are best (highest reward) in each row. The $\pm$ denotes standard deviation.

| Task | IQL (Oracle) | PT | PT+ADT | DPPO | SARA |
|---|---|---|---|---|---|
| pen-human-v1 | 85.19 ($\pm$62.1) | 80.84 ($\pm$63.1) | **82.80** ($\pm$**62.4**) | 76.97 ($\pm$64.3) | 81.89 ($\pm$63.2) |
| pen-cloned-v1 | 81.49 ($\pm$62.6) | 70.28 ($\pm$64.6) | 69.70 ($\pm$63.8) | **72.13** ($\pm$**64.5**) | 69.72 ($\pm$63.4) |

Table 12: Average normalized policy evaluation rewards (across 8 seeds), using **human-labeled preference data without neutral preferences**. Values in **bold** are best (highest reward) in each row and underlined are within 1% of best. The $\pm$ denotes standard deviation.

| Task | IQL (Oracle) | PT | PT+ADT | DPPO | SARA |
|---|---|---|---|---|---|
| pen-human-v1 | 85.19 ($\pm$62.1) | 83.59 ($\pm$62.8) | **85.06** ($\pm$**61.7**) | 75.51 ($\pm$63.8) | 81.52 ($\pm$62.4) |
| pen-cloned-v1 | 81.49 ($\pm$62.6) | 70.18 ($\pm$64.0) | 69.70 ($\pm$64.3) | 70.95 ($\pm$65.6) | **71.07** ($\pm$**64.3**) |

Table 13: Average normalized policy evaluation rewards (across 8 seeds), using **script-labeled preference data**. Values in **bold** are best (highest reward) in each row and underlined are within 1% of best. The $\pm$ denotes standard deviation.

| Task | IQL (Oracle) | PT | PT+ADT | DPPO | SARA |
|---|---|---|---|---|---|
| pen-human-v1 | 85.19 ($\pm$62.1) | 81.67 ($\pm$62.8) | 82.80 ($\pm$63.8) | 71.62 ($\pm$66.0) | **83.06** ($\pm$**63.0**) |
| pen-cloned-v1 | 81.49 ($\pm$62.6) | **76.88** ($\pm$**65.3**) | 74.30 ($\pm$65.3) | 75.41 ($\pm$64.6) | 73.38 ($\pm$65.6) |

infer rewards using this encoder learned from the hopper preferences. This cross-task transfer of preferences enables policy learning with these estimated rewards (Figure 3). Remarkably, the IQL learned policy from cross-task preferences performs only slightly worse than the SARA model using the walker replay preference set. Both exhibit lower variance than using the true task reward.

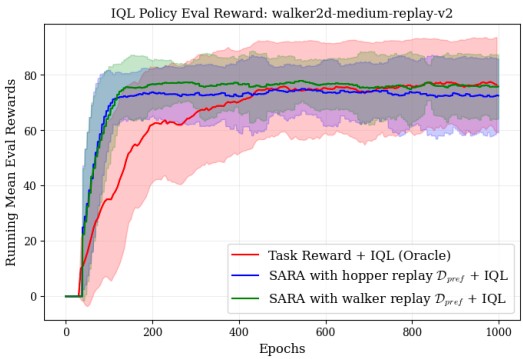

Figure 3: Walker2d IQL eval rewards, running average and 8 seeds (see Section G.2). Shading indicates standard deviation. The policy is learned on the walker replay dataset. SARA rewards from training on hopper replay preferences perform almost as well as SARA rewards with training on the walker replay preferences.

Hejna et al. (2020) investigated transfer learning of a policy from a simple agent to a more complex one with environment provided rewards, not preference aligned rewards. Liu et al. (2024) proposed an optimal transport method to transfer preferences, but their framework is limited to tasks with equivalent state-action space. To our knowledge, our cross-task preference transfer to a larger state-action space is novel, and we do so without any changes to the SARA architecture or hyperparameters.

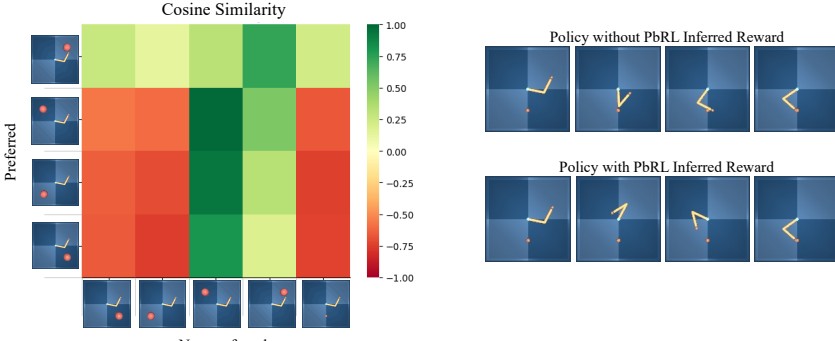

Figure 4: Left: After training SARA encoder, we divide up the trajectories by their label (preferred vs non-preferred) and by the target location. Note that the training set has the extra hard target location for the non-preferred but not preferred. Target location is not explicitly fed into the model and many trajectories are the same between the two sets. Right: With task reward alone, the learned policy takes the shortest path going clockwise to the hard target location. Including the preference inferred award enables a learned policy traversing in counter-clockwise direction.

**Online RL with reward shaping**  In contrast to previous sections, we now consider the scenario of some known but under-specified task reward. For instance, consider a robotics application in which a policy is trained to achieve some task, such as reaching an object. In many realistic applications, such task-driven reward functions can result in policies that have undesirable patterns of movement, such as rapid or jerky actions (Escontrela et al., 2022). Such movements may not only be visually unappealing but can also damage the robot. Prior works employed complicated reward shaping techniques to overcome these issues (Miki et al., 2022). Here we test whether the SARA inferred rewards can shape the task reward in online RL learning.

We set up the following problem using the Deepmind Control Suite Reacher task (Tunyasuvunakool et al., 2020; Laskin et al., 2021). Let us assume a human labeler decided that counterclockwise movements are always preferable, perhaps due to a realistic engineering constraint or potential environmental obstructions limiting clockwise movement. The human labeler picks the preferred

set of trajectories (all counterclockwise) for an easy target. The human labeler deems non-preferred any trajectories from the policy trained on the task reward. These are sometimes clockwise and sometimes counterclockwise depending on which has the fewest steps to target. Therefore, when the target is in the upper half, we have many trajectories for which the preferred and non-preferred sets overlap. This setup requires SARA to disentangle the preferred from non-preferred styles even when there are some strong similarities between the two groups.

The encoder is not explicitly given the counter-clockwise preference, so we evaluate what it implicitly learns by comparing with non-preferred movements. After training we split up trajectories by category (preferred vs non-preferred) and target location to view the similarity map shown in Figure 4. Despite overlaps between categories, SARA learns where preferred and non-preferred behaviors align or diverge. This result arises from the subset contrastive encoding and shuffling detailed in Section 3.

Next we test whether these learned patterns transfer to the harder task of the small target shown in Figure 4. We learn online using the Deep Deterministic Policy Gradient algorithm (Lillicrap et al., 2016). We add the known task reward to the SARA inferred reward. In the absence of our SARA inferred reward, the learned Reacher policy takes the most efficient path clockwise to the hard target (Figure 4). With our preference inferred reward, the policy takes the desired counterclockwise path even though it results in a lower task driven reward. The SARA framework achieves this even though the preferred set only includes the easy large target task.

This toy experiment illustrates our model's potential for the realistic preference driven goal of shaping an RL policy to conform with human desires. Such stylistic rewards may be both application specific and difficult to engineer, so inferring from preference data is a promising path forward.

## D  DATA SCALING EXPERIMENTS

Here we investigate whether SARA requires more data than BT based preference modeling to achieve its robustness advantage. We ran SARA and the Bradley-Terry based Preference Transformer (PT) using only *half* the originally given preference data. For these reduced preference datasets, we ran both 0% error rate and 20% error rate. At 0% error rate, we note SARA is less performant compared to PT on multiple datasets. However, at 20% error rate, SARA substantially outperforms PT on multiple datasets. Therefore, SARA still offers a robustness advantage over the Bradley-Terry based PT model in the low data regime.

Table 14: Average normalized policy evaluation rewards (8 seeds) using 50% of preference datasets (0% error rate). Values in **bold** are the highest per row. The $\pm$ denotes standard deviation.

| Task | PT | SARA |
|---|---|---|
| hop-medium-replay | 83.59 $\pm$14.3 | **85.00** $\pm$2.6 |
| hop-medium-expert | **82.20** $\pm$33.8 | 53.90 $\pm$49.8 |
| walk-medium-replay | 75.33 $\pm$4.0 | **76.98** $\pm$6.9 |
| walk-medium-expert | **109.59** $\pm$1.3 | 107.68 $\pm$7.0 |
| kitchen-partial | **59.96** $\pm$14.4 | 56.13 $\pm$22.0 |
| kitchen-mixed | **54.38** $\pm$10.0 | 51.29 $\pm$10.8 |

## E  DATASET DETAILS

Here we describe the details of the preference datasets and the full offline datasets used for our offline RL experiments (Section 4)

### E.1  PREFERENCE DATASETS

For Mujoco locomotion and Adroit pen tasks, we use the preference datasets provided by Kim et al. (2023) from the PT human label repository. For the Franka Kitchen tasks, we use the preference

Table 15: Average normalized policy evaluation rewards (8 seeds) using 50% of preference datasets (20% error rate). Values in **bold** are the highest per row. The $\pm$ denotes standard deviation.

| Task | PT | SARA |
|---|---|---|
| hop-medium-replay | 54.99 $\pm$28.2 | **84.47** $\pm$3.1 |
| hop-medium-expert | 63.00 $\pm$40.2 | **63.48** $\pm$43.1 |
| walk-medium-replay | 71.55 $\pm$10.1 | **74.28** $\pm$10.8 |
| walk-medium-expert | **108.56** $\pm$4.8 | 108.39 $\pm$4.7 |
| kitchen-partial | 39.26 $\pm$26.6 | **41.95** $\pm$25.8 |
| kitchen-mixed | 41.99 $\pm$16.3 | **48.16** $\pm$15.3 |

datasets from An et al. (2023) from the DPPO human label repository. Both repositories are MIT licensed.

Kim et al. (2023) and An et al. (2023) created the preference datasets by sampling pairs of trajectories from the full offline datasets Fu et al. (2021). They named the preference datasets by the same name as the full offline datasets (e.g. hopper-medium-replay-v2 and so on). All trajectories are 100 timesteps in length. The replay datasets for hopper and walker have 500 trajectory pairs and all others have 100 pairs. The labelers are domain experts who are given specific criteria upon which to evaluate their preference for the trajectories. We refer the reader to the original works which state their preference criteria. Excluding pairs with equally preferred trajectories is one of our dataset variants, so here we provide the number of such queries in each dataset (Table 16.

Table 16: Percentage of neutral queries by preference set.

| Preference Set | Total Pairs | Neutral (%) |
|---|---|---|
| hopper-medium-replay-v2 | 500 | 38% |
| hopper-medium-expert-v2 | 100 | 28% |
| walker2d-medium-replay-v2 | 500 | 23% |
| walker2d-medium-expert-v2 | 100 | 24% |
| kitchen-partial-v0 | 100 | 22% |
| kitchen-mixed-v0 | 100 | 24% |
| pen-human-v1 | 100 | 35% |
| pen-cloned-v1 | 100 | 40% |

### E.2 FULL OFFLINE DATASETS

In our offline RL experiments, the policies for the SARA framework, all baselines, and the oracle are trained using the full offline D4RL datasets. The oracle uses the true environmental rewards provided in the datasets. In the case of the SARA framework and baseline models, each transition reward is computed using the respective models. Note all these models are non-Markovian, so each transition reward at time $t$ is computed by feeding the trajectory up to and including time $t$ into the models. For each model, we replace the dataset provided rewards with the computed transition rewards.

We refer to the work by Fu et al. (2021) for a thorough description of the full offline datasets. Here we summarize some key points as they relate to our work. Our experiments include hopper and walker2d locomotion tasks from Gym-Mujoco. The hopper has a 3 dimensional action space and 11 dimensional state space. The walker2d has a 6 dimensional action space and 17 dimensional state space. Franka Kitchen tasks are multi-task and high dimensional, requiring algorithms to "stitch" trajectories. The action space is 9 dimensional and the state space is 60 dimensional. The Adroit pen tasks are also high dimensional and contain a narrow distribution of expert or cloned expert data. The action space is 24-dimensional and the state space is 55-dimensional. By testing the methods on the 8 datasets from these three environments, we experiment on a range of task dimensionalities and difficulties.

## F SARA FRAMEWORK ARCHITECTURE AND HYPERPARAMETERS

First we provide the architecture of our contrastive encoder. Then we provide the hyperparameters used for acquiring the results in our main paper.

### F.1 TRANSFORMER ARCHITECTURE

Here we briefly summarize architecture and hyperparameters for our contrastive encoder. Both Transformer Encoders use the standard Pytorch implementation pyt, which is based upon the originally proposed Transformer architecture by Vaswani et al. (2017). The Transformer Encoder 1 encodes each trajectory (state and action sequences), but it does not enable attention between trajectories. Information at different timesteps within each trajectory can attend to one another. We inject temporal information via positional encoding Vaswani et al. (2017). We experimented with causal masking in the encoder training, where state-actions can only attend to previous state-actions but not future. However, we found that this masking made either no difference or slightly degraded performance in downstream IQL learning.

We conduct average pooling over timesteps for each trajectory, resulting in one latent per trajectory: $\mathbf{u}_{p/n,i}$, where $p$ or $n$ indicates preferred or non-preferred and $i$ indexes trajectory. Next we form $k$ subsets within each category, *i.e.* $k$ subsets within the set of $\{\mathbf{u}_{p,i}\}$ and k subsets within the set of $\{\mathbf{u}_{n,i}\}$. Each subset is comprised solely of trajectories for either preferred or non-preferred. Then we pass each subset $\{\mathbf{u}_p\}_k$ and $\{\mathbf{u}_n\}_k$ through Transformer Encoder 2, enabling encodings in the same subset to attend to each other. Note the time dimension was already removed prior to this encoder, so we do not have any positional encoding here. We then have single encoding $\mathbf{z}_{p/n,k}$ for each subset. This is trained with the SimCLR loss with a temperature hyperparameter Chen et al. (2020). The SimCLR loss does the following: pulls together latents within the same type (p or n) and repels each of the $\{\mathbf{z}_{p,k}\}$ from each of the $\{\mathbf{z}_{n,k}\}$. As noted in Section 3, we shuffle the composition of latents in each subset $\{\mathbf{u}_p\}_k$ and $\{\mathbf{u}_n\}_k$ to ensure robustness to mislabeling or existence of similar trajectories in the two sets.

As noted in the main paper, we conduct each experiments over 8 seeds. The seeding not only applies to the downstream IQL training but also the encoder training. We do this to align with our baseline models Kim et al. (2023); An et al. (2023), which also seed their Preference Transformer and DPPO Probability Predictor, respectively, with the same seed as their downstream policy training.

### F.2 SARA ENCODER HYPERPARAMETERS

Unless otherwise noted, all hyperparameters were kept the same for all preference datasets, even though the Kitchen and Adroit environments have higher dimensional action/state spaces than the locomotion tasks.

Here we provide hyperparameters for both Transformer Encoder 1 and Transformer Encoder 2

Table 17: Transformer Encoder 1 and 2 Hyperparameters

| Hyperparameter | Value |
|---|---|
| Causal pooling | No |
| Model dimension ($d_{\mathrm{model}}$) | 256 |
| Feedforward network dimension | 256 |
| Embedding dimension ($\mathbf{z}_{p/n}$ dim) | 16 |
| Encoder dropout rate | 0.0 |
| Positional encoding dropout rate | 0.0 |
| Number of encoder layers | 1 |
| Number of attention heads | 4 |
| Avg pooling (after 1st encoder) | Yes |

Here we provide additional training hyperparameters:

Table 18: Training Hyperparameters

| Hyperparameter | Value |
|---|---|
| Batch size | 256 |
| Use cosine learning-rate schedule | Yes |
| Initial learning rate | $1 \times 10^{-5}$ |
| Min learning rate | $1 \times 10^{-6}$ |
| Optimizer | Adam |
| Number of epochs | 2000 (hopper expert), $10^4$ (walker replay), 4000 (all others) |
| Sets per category ($k$) | 2 |
| Temperature (SimCLR loss) | 0.1 |

The sequence lengths in the preference sets are all 100. As done by the DDPO baseline An et al. (2023), we experimented with using subsequences of varying lengths in training. We passed in subsequence lengths of $[10, 20, 30, 40, 50, 60, 70, 80, 90, 100]$, and we used random start points in the sequences. We found that using random subsequences to train the encoder resulted in slightly reduced variance in the downstream IQL training on the hopper replay dataset. However, in general the asymptotic performance in IQL training was not sensitive to whether or not we used random subsequences of varying length.

# G   POLICY TRAINING AND EVALUATION

We first provide details regarding policy training. We then detail the evaluation method.

## G.1   POLICY TRAINING HYPERPARAMETERS

We train policies for oracle, SARA, PT, and PT+ADT using the IQL implementation in the publicly available OfflineRL-kit Sun (2023). We carefully match hyperparameters to those suggested by Kim et al. (2023), which also match the hyperparameters suggested for the offline datasets in Kostrikov et al. (2022). In these works, the Gym-Mujoco environments have different IQL hyperparameters, namely for dropout and temperature, than the ones used for the Franka Kitchen and Adroit environments. We also use the same reward normalization functions provided byKim et al. (2023). We also carefully match hyperparameters for DPPO policy training to the ones provided by An et al. (2023) in their appendices and code repository. We defer to Kim et al. (2023) and An et al. (2023) for exact hyperparameters. Upon acceptance, we will release our IQL training pipeline with the hyperparameters used for each offline dataset.

## G.2   POLICY EVALUATION METHOD

To compare methods, we roll out multiple evaluation episodes for the method's learned policy and get the normalized trajectory reward provided by the environment, as done in prior PbRL works. We use the $get\_normalized\_score$ functions provided by each environment, which uses scaling factors unique to that environment. We scale the episode returns by 100 as done in prior works.

To avoid reporting overly optimistic values, we follow the method proposed by Hejna et al. (2024). We roll out 10 evaluation episodes every 5 epochs of training. We compute the average and the standard deviation of the true normalized episode rewards over the last 8 evaluations. Thus 80 total evaluation episode rewards are averaged at each epoch. This running mean is averaged over the 8 seeds. We report the maximum value achieved after averaging the running mean over the seeds. As noted in Hejna et al. (2024), this maximum of seed-averaged running mean mitigates effects of stochasticity. Past works either do not provide details on the metric computation or report the seed-averaged maximum, which can inflate performance. To report standard deviations that capture both within-seed and across-seed variability, we compute the total standard deviation as follows.

At each epoch of training a particular seed $s \in \{1, \ldots, S\}$, we have a set of $n = 80$ evaluation episodes. For each seed, we compute the standard deviation over episodes and apply Bessel's

correction:

$$\sigma_s^{\text{corrected}} = \sigma_s \cdot \sqrt{\frac{n}{n-1}}.$$

The *within-seed variance* is the average of squared corrected standard deviations:

$$\sigma_{\text{within}}^2 = \frac{1}{S} \sum_{s=1}^{S} \left( \sigma_s^{\text{corrected}} \right)^2.$$

Let $\mu_s$ denote the mean return for seed $s$. The *across-seed variance* is the unbiased sample variance of the seed means:

$$\sigma_{\text{across}}^2 = \frac{1}{S-1} \sum_{s=1}^{S} \left( \mu_s - \bar{\mu} \right)^2, \quad \bar{\mu} = \frac{1}{S} \sum_{s=1}^{S} \mu_s.$$

The total standard deviation used for error bars is then given by:

$$\sigma_{\text{total}} = \sqrt{\sigma_{\text{within}}^2 + \sigma_{\text{across}}^2}.$$

We use the same seeds and reporting methods for all reported values, including for SARA, baselines, and the oracle.

## H  CROSS-TASK TRANSFER OF PREFERENCES

Table 19: Hopper to walker2d action and observation dim mapping

| **Action Mapping** | | |
|---|---|---|
| **hopper dims** | **walker2d dims** | **actions in walker2d** |
| 0–2 | 0–2 | torques on thigh, leg, foot joints (right) |
| 0–2 | 3–5 | torques on thigh, leg, foot joints (left) |
| **Observations Mapping** | | |
| **hopper dims** | **walker2d dims** | **observations in walker2d** |
| 0–1 | 0–1 | height and angle of top of torso |
| 2-4 | 2-4 | angle of thigh, leg, foot joints (right) |
| 2–4 | 5–7 | angle of thigh, leg, foot joints (left) |
| 5–7 | 8–10 | velocity of x coordinate, height coordinate, and angular velocity of top |
| 8–10 | 11–13 | angular velocities of thigh, leg, foot hinges (right) |
| 8–10 | 14–16 | angular velocities of thigh, leg, foot hinges (left) |

As discussed in Section C we train on the hopper-medium-replay-v2 preference set, and we use the learned preferred latent to compute rewards for the full offline walker2d-medium-replay-v2 dataset. We then conduct IQL training on this walker2d dataset. In order to accomplish the SARA reward computation we must build an encoder based on the hopper data that can accept walker state-action space dimensions. The online Gym documentation provides a detailed description of the hopper and walker2d state and action spaces hop (a); wal. We need to map the 3-dimensional hopper action space to the 6-dimensional walker2d action space. We also need to map the 11-dimensional hopper state space to the 17-dimensional walker2d state space. To do so we exploit the symmetries in the walker joints as shown in Table 19.

Of course this is not a physically realistic way to map the dimensions. In a well trained walker2d policy, the two legs are not moving symmetrically. Nonetheless, we train an encoder on the hopper replay data with dimensions mapped in this way, and subsequently we infer a preferred latent with this modified hopper replay data. Next, we take the full offline D4RL walker2d replay dataset, and we pass each trajectory through the encoder to get latents for each timestep in each trajectory. Next

we compute rewards for each timestep in each trajectory in the walker2d replay dataset by computing cosine similarity with the preferred latent. Lastly we conduct IQL training and evaluate as we did with all other datasets. Though we based this method on physically unrealistic assumptions, we acquire normalized policy rewards that are only few points worse than the reward values attained using the walker2d replay preference set (Figure 3). We also result in lower evaluation reward variance compared to the oracle.

## I    ANTMAZE BUG

In their DPPO paper, An et al. (2023) found a critical bug in the Antmaze environment's goal randomization (Appendix F of original paper). After fixing the bug, the authors showed that state-of-the-art offline RL algorithms acquire trivially low policy returns (¡12) even with the true environmental rewards. Therefore, we align with An et al. (2023) by deferring experiments on Antmaze until the offline-RL community can investigate further.

## J    COMPUTE RESOURCES

Our experiments involved training the following individual models on multiple seeds and datasets: SARA contrastive encoder, PT, PT+ADT, IQL policy training, DPPO preference predictor, and DPPO policy. Each individual model was trained on a single NVIDIA A100-SXM4-80GB GPU and 16 CPU cores. Compute resources needed were less than 20GB GPU per model. Training time for the SARA encoder, PT, PT+ADT, and DPPO preference predictor varies depending on size of dataset and number of epochs, but it was typically under 30 minutes per model. The walker2d-medium-replay-v2 dataset took up to 2 hours to train a SARA encoder for 10000 epochs when using additional random slices of trajectories. The DPPO policy training took approximately 2 hours to train on one model on the single GPU. The IQL policy training, using the open source OfflineRL-kit Sun (2023), took about 3.5 hours to train one model. These computing resources were used for 8 datasets, each model, and 8 seeds per model+dataset. We used our university's computing cluster with access to multiple GPUs, but the models could also be trained on a single standalone GPU or with increased training time, on CPUs alone.

## K    LLM USAGE

LLM tools were used in this research and paper in the following manner. The authors used code snippets to generate plots and making LaTex tables. LLMs were also used in refining small portions of text in paper writing. In response to specific prompts, LLMs were used for generating helper functions in pipeline code. LLMs were **NOT** used for writing large portions of the paper or generating original ideas Therefore, an LLM is not a significant contributor or author of this work.

