# OpenReview forum: "Similarity as Reward Alignment: Robust and Versatile Preference-based Reinforcement Learning"
_ICLR.cc/2026/Conference — Submitted to ICLR 2026_

### Official Review · Reviewer_y4XU · 2025-10-28

**Soundness:** 2
**Presentation:** 1
**Contribution:** 2
**Rating:** 2
**Confidence:** 3

**Summary:**

The paper proposes a preference-based reinforcement learning method named SARA, which aims to construct latent representations of preferred trajectories via contrastive learning and compute rewards based on similarity to enhance robustness against noisy labels.

**Strengths:**

1) The paper validates the proposed framework through extensive experiments across multiple environments, demonstrating its effectiveness in improving performance.

2) Additional robustness experiments are conducted, verifying the algorithm's resistance to noise to some extent.

**Weaknesses:**

1) The underlying mechanism and motivation for using similarity as reward are unclear.

2) The innovation is limited, primarily involving integration and improvements of existing frameworks, with little theoretical analysis.

3) The writing is generally verbose, and the overall quality still falls short of ICLR standards.

**Questions:**

Please see weaknesses.

---

> ### Author Response · Authors · 2025-11-21
> **Response**
>
> Thank you for your time in reading our paper and providing your review. As you noted, we did conduct extensive experiments to validate our model. We would like to address your concerns as well as pose some follow up questions.
>
> **Weakness 1: Underlying motivation for using similarity as reward**
>
> We had detailed our theoretical justification for the similarity as reward metric in Appendix A, but we agree that it is clearer to integrate this explanation into the main paper. In our uploaded revision, we expanded Section 3 (pages 4-5) to include the justification.
>
> Here we provide an overview of the motivation that we present in our paper. The primary goal of SARA is robustness to labeler error.  To this end, we first learn an error robust representation of the preferred set, and we want our reward metric to follow from this robust representation. We use the SimCLR objective to learn this representation; this objective explicitly optimizes cosine similarity between representations of preferred subsets. Thus, using cosine similarity as the reward metric follows naturally: trajectories aligned with the preferred direction receive higher reward, and those misaligned receive lower reward.
>
> **Weakness 2: Limited innovation and theoretical analysis**
>
> Our primary technical innovations are the following. First we learn a representation of the **set** of preferred trajectories, and then we use a **cosine similarity reward** metric following naturally from our encoder learning objective.
>
> In contrast to the majority of prior work, we do not use the Bradley Terry model, and this modeling choice is in itself a large deviation from prior PbRL literature. Most prior works differ from our work in two ways: 1) they learn representations at the **individual** trajectory level and/or 2) a reward model comparing **individual** trajectories using the **Bradley-Terry model**. Our mechanistic analysis (Section 5, page 8) shows that these prior frameworks suffer from direct corruption of the loss gradient in the presence of label noise.
>
> We are not aware of a method in the PbRL literature that: (a) embeds an entire set of preferred trajectories as a unit or (b) define rewards directly via cosine-similarity to the preferred set embedding. If there are prior PbRL works that use our same innovations, we would appreciate the opportunity to compare SARA to these works and cite them in our paper. Could you please provide references to these works?
>
> **Weakness 3: writing and quality**
>
> Could you please provide specific feedback on the writing style and quality? The other reviewers rated the presentation and overall scores more favorably; on details where they had questions or lacked clarity, they noted their specific concerns. We would like to address your concerns on quality and style, but we are unable to do so without concrete details of how style and quality fall short.

---

### Official Review · Reviewer_upAd · 2025-10-30

**Soundness:** 2
**Presentation:** 2
**Contribution:** 2
**Rating:** 4
**Confidence:** 4

**Summary:**

This paper studies the problem that most PbRL methods are fragile when human feedback has noise, which often happens with non-expert labelers. The authors propose SARA, which learns a latent space of preferred samples and uses similarity in this space as the reward instead of directly predicting a reward from noisy labels.

**Strengths:**

This paper proposes a simple method that makes PbRL more robust when preference labels contain noise. The approach is practical in real settings where human annotators often make mistakes.

**Weaknesses:**

The experimental evaluation is limited, as standard offline PbRL studies usually include a broader set of tasks. In addition, some more recent baselines are missing. These gaps in the experimental design raise concerns about the reliability and generality of the reported performance.

**Questions:**

1.The legend in Figure 2 is very cluttered, making it hard to understand the intended message from the plot.

2.Is the method robust when preferences come from multiple annotators? Different users may have different preference patterns, leading to conflicts beyond simple label flips. Can SARA handle such heterogeneous preference distributions?

3.Why is the proposed method more robust to label noise than existing approaches based on the BradleyTerry model? A clearer theoretical or empirical justification would be helpful.

4.The paper mentions assumptions behind SARA. Can the authors analyze in which scenarios SARA may fail, or discuss the limits of its applicability? Understanding the boundary conditions would strengthen the work.

5.Since SARA learns trajectory encoding in a latent space, what happens if preferred trajectories are multi-modal? Will the latent representation collapse or fail to capture diverse preference modes?

---

> ### Author Response · Authors · 2025-11-21
> **Response**
>
> Thank you for your time in reading our paper and for your constructive response. We appreciate you noting the utility of our approach in the realistic setting where annotators make mistakes. Here we address your comments and questions in detail, and we would appreciate any further response.
>
> **Weakness 1: additional tasks and baselines**
>
> We agree that our work would be strengthened by inclusion of additional tasks and baselines. We ran the model on Halfcheetah using human labeled preference datasets provided by An et al. Below are the the average normalized policy evaluation rewards (across 8 seeds). Highest values are bold. The +- denotes standard deviation. The models perform similarly and SARA provides competitive results against baselines.
>
>  **0% error rate**
>
> | Task | PT | PT+ADT | DPPO | SARA |
> | --- | --- | --- | --- | --- |
> | halfcheetah-medium-replay | 40.94 ± 2.7 | **42.85 ± 1.7** | 39.94 ± 4.3 | 41.65 ± 2.0 |
> | halfcheetah-medium-expert | 86.62 ± 14.2 | 89.46 ± 9.4 | **92.18 ± 8.5** | 86.56 ± 13.5 |
>
> **20% error rate**
>
> | Task | PT | PT+ADT | DPPO | SARA |
> | --- | --- | --- | --- | --- |
> | halfcheetah-medium-replay | 41.16 ± 1.9 | **42.25 ± 2.2** | 38.59 ± 6.9 | **42.08 ± 2.1** |
> | halfcheetah-medium-expert | 87.97 ± 11.1 | 89.18 ± 10.4 | **92.32 ± 7.6** | 88.41 ± 10.4 |
>
> We are also running additional tasks, and we will post the results as soon as they are finished. We would also like to note that compared to prior PbRL works, we aimed to provide a wider breadth of analysis to include varying error rates, Pearson correlation to environmental rewards, and an online experiment with non-paired feedback.  Most previous offline PbRL works focus on evaluation returns in the offline setting with expert labeling, without error. By contrast, we provide results for error rates of 0, 10, 20, and 40%. We also show results in excluding neutral responses as well as script labeling by an oracle (Tables 7-9). Prior PbRL works often do not include any analysis of correlation to the environment provided rewards, while ours does so under these varying error rates (Table 3 and 6).  Lastly, prior works largely focus on narrow domains (offline RL for example) whereas we apply our model to an online RL experiment with non-paired feedback format and cross-task preference transfer (Appendix C). We believe our breadth of results indicates promise in PbRL experiments with label noise.
>
> **Question 1: Figure 2 intended message**
>
> Thank you for noting interpretability challenges in our Figure 2. To clarify the intended message, we replaced Figure 2 with the following table in our revision.
>
> Our paper shows results for six labeling conditions (Tables 7-9): four error rates, script labeling by oracle, or exclusion of neutral queries. Here, we show the mean normalized episode evaluation returns across the six labeling conditions. The +- is the standard deviation across the six labeling conditions. We bold the **highest mean** and **lowest standard deviation** in each row. While baselines may win in mean or standard deviation as one-offs, SARA is the method that wins most frequently in highest mean and lowest variance across label conditions. In cases where SARA loses compared to baselines, the margin by which it loses is quite small. We believe this evidences SARA’s consistency in response to varying label quality.
>
> | Task | PT | PT+ADT | DPPO | SARA |
> | --- | --- | --- | --- | --- |
> | hopper-med-replay | 65.73 ± 25.49 | 71.29 ± 24.02 | 57.31 ± 26.71 | **81.49** ± **14.64** |
> | hopper-med-expert | 68.73 ± 36.17 | 77.09 ± **33.96** | 83.92 ± 36.05 | **85.22** ± 34.68 |
> | walker2d-med-replay | 72.35 ± 12.65 | 72.58 ± 16.25 | 43.93 ± 28.13 | **75.95** ± **11.29** |
> | walker2d-med-expert | 107.10 ± 9.41 | 106.42 ± 10.82 | **108.73** ± **0.42** | 108.70 ± 4.57 |
> | kitchen-partial | 59.69 ± 17.90 | 60.76 ± **17.06** | 38.50 ± 18.51 | **61.01** ± 17.93 |
> | kitchen-mixed | **49.85** ± 15.53 | 48.57 ± 16.21 | 45.68 ± 17.97 | 49.03 ± **11.40** |
>
> Note that for within a labeling condition there is variance arising due to the 8 seeds. The reported standard deviations in this table accounts for both sources of variance (across the labeling condition and within labeling condition).

---

> ### Author Response · Authors · 2025-11-21
> **Response continued**
>
> **Question 2: Heterogenous preferences among annotators**
>
> The annotators are explicitly instructed to follow a given *single* *specifc preference criteria* (e.g., “hopper: walk smoothly and quickly without tripping or falling”). Thus, we and prior baseline approaches assume the problem setup of uni-modal preference criteria. The non-preferred trajectories should capture all other patterns (slow and steady, slow and unstable, fast and unstable, etc). Under the given preference criteria, annotators who prefer a pattern such as slower motion are violating the instruction; their annotations then fall under labeler error.
>
> Although the paper is focused on the single criteria setup, SARA can be used for multi-preference criteria. For our problem setup, we assumed two subset categories for Transformer Encoder 2, preferred and non-preferred. With multiple criteria, such as “preferred-fast”, “preferred-slow”, “non-preferred”,  we would now have three categories which we train contrastively. However, we have not tested this due to lack of a multi-preference dataset for RL environments.
>
> In summary,  1) the scope of this work and our baselines is uni-modal preference criteria; 2) if annotators label in accordance with preferences that fall outside of a single strict preference criteria, then this amounts to label error; 3) SARA’s architecture can be straightforwardly used for multi-modal preference criteria.
>
> **Question 3: Robustness to label noise compared to the BT model**
>
> Thank you for the question. Here we attempt to clarify our Section 5 mechanistic justification.
>
> The key structural difference is that BT directly models **individual** trajectory comparisons. Each mislabeled pair $(\sigma_i, \sigma_j)$ corrupts the BT gradient in proportion to
>
> $\nabla_r \log P[\sigma_j \succ \sigma_i; r] = -\gamma \nabla_r \log P[\sigma_i \succ \sigma_j; r]$. The coefficient $\gamma=\frac{\text{sigmoid}(r(\sigma_i)-r(\sigma_j))}{\text{sigmoid}(-(r(\sigma_i)-r(\sigma_j)))}$ is always positive and blows up for large reward differences.
>
> By contrast, SARA does not model individual comparisons. Its **set-based** contrastive objective encourages the encoder to learn **aggregate patterns** that consistently distinguish the preferred set from the non-preferred set.  Even if some trajectories in the preferred set are mislabeled, the encoder must still produce a set representation $z_p$ that is dissimilar from the non-preferred set. To do so, the contrastive objective drives the following:
>
> 1. **Encoder 1** places low attention weight on portions of mislabeled trajectories that resemble the non-preferred set. It focuses instead on transitions shared by the majority of correctly labeled preferred trajectories.
> 2. **Encoder 2** aggregates these per-trajectory latents into a consensus representation that maximally separates the two sets.
>
> In this way, the contrastive objective naturally aggregates to a dominant structure in the preferred set, rather than directly following gradients of individual noisy labels as BT does. Even when mislabeling occurs, the resulting $z_p^*$ is still a latent that shows low similarity to the non-preferred sets and high similarity to subsets of preferred.
>
> Finally, we note the transformer architecture induces complex nonlinear interactions, both between timesteps within a trajectory and across trajectories within each set. Thus, a formal robustness proof is currently out of reach. As with prior PbRL methods (including our BT-based baselines) and much of transformer-driven empirical research, we rely on intuitive mechanistic reasoning and the empirical validation presented in the paper.

---

> ### Author Response · Authors · 2025-11-21
> **Response continued**
>
> **Question 4: limits of SARA applicability**
>
> Thank you for this thoughtful question. We showed in our paper does not always offer a clear advantage over some baseline models on medium-expert datasets, although it is generally competitive against these models. On these datasets, BT-based models on pairs of trajectories, where the expert and medium trajectories differ sharply, can still learn a reasonable reward signal. In contrast, the medium-replay datasets contain more varying quality data, making them less robust to noise when learning on pairs of individual trajectories. Our result is consistent with prior work (Sun et al. 2025), which shows that BT models can perform well when annotation quality is high.
>
> **Question 5: capturing diverse preference modes**
>
> We interpreted question 2 to refer to annotators having varying preference criteria.  By contrast, here we assume the question is regarding a single preference criteria, but there are multi-modal behaviors which align equally well to the single criteria. This is also an important consideration.
>
> A full-fledged analysis would require additional human preference labeled datasets that exhibit such multi-modality, but we see no reason to expect representation collapse. Just as a transformer trained on diverse language does not simply average distinct tokens into a single mode, we expect that the SARA encoder, tuned to sufficient capacity, can also represent multi-modal behavior. The SARA encoder learns to **attend** to trajectories that most clearly distinguish from the non-preferred set. Therefore, the $z_p^*$ is **not a mean** over the preferred trajectory latents but an attention-weighted representation that reflects the most discriminative features.
>
> Consider a 2d gridworld task to reach a goal above with a barrier in the way. There are two optimal paths from start to goal in the same number of steps: 1) left and up or 2) right and up. Both trajectories would get lumped into preferred sets. Trajectories going away from the goal (ie down) or up into the barrier would be non-preferred. In this case, reward with respect to $z_p^*$ would incentivize movement towards to goal in either left/up or right/up, but it does not represent an average of the two to incentivize directly up into the barrier. Any movement attempting to go directly up into the barrier or away from the goal would be in the non-preferred set, and these behaviors would be disincentivized.

---

### Official Review · Reviewer_LfAy · 2025-10-31

**Soundness:** 3
**Presentation:** 3
**Contribution:** 3
**Rating:** 6
**Confidence:** 3

**Summary:**

SARA (Similarity as Reward Alignment) replaces Bradley-Terry reward modeling in preference-based RL with a contrastive learning approach. It yields robust rewards under noisy labels and supports cross-task preference transfer and reward shaping.

**Strengths:**

- Robustness: Maintains stable performance even with high label noise (10–40%).
- Simplicity: No Bradley–Terry modeling or pairwise loss; uses a single contrastive objective.
- Empirical Performance: Outperforms or matches strong PbRL baselines across D4RL benchmarks.

**Weaknesses:**

- The fixed prototype $z_p^*$ might drift if new or biased preference data are added; continual adaptation requires full retraining.
- There is no analysis of how performance scales with the number of preference pairs or trajectories. We don’t know whether SARA needs more data than BT to learn a stable prototype, or how small-sample performance behaves.

**Questions:**

- How is $z_p^*$ updated when new preferences are added or when different annotator populations are used? Does it require retraining from scratch?
- How does SARA’s performance change with fewer preference samples? Does it require more data than BT to stabilize the pivot latent?

---

> ### Author Response · Authors · 2025-11-21
> **Response**
>
> Thank you for your time in reading our paper. We appreciate you noted our model’s robustness, simplicity, and strong empirical performance. We hope to address your questions, and we invite further response.
>
> **Weakness 1 and Question 1: re-training for new preference data**
>
>  Thank you for noting this important research direction. We have not yet explored efficient fine-tuning of SARA as new preference data are added.  Our models and baseline models assume a fixed preference dataset. We agree that methods of efficiently updating SARA in response to new preference data should be explored, but it is outside the scope of the problem setup for our model and our baselines. Our Limitations and Future Work section suggests adaptation of SARA for such scenarios.
>
> **Question 1: different annotator populations**
>
> The second part of question 1 asks about training with different annotator populations. We would like to clarify that annotators are explicitly given one specific preference criteria for each dataset (e.g., “hopper: walk smoothly and quickly without tripping or falling”). Thus, we and prior baseline approaches assume the problem setup of uni-modal preference criteria. The non-preferred trajectories should capture all other patterns (slow and steady, slow and unstable, fast and unstable, etc). Under the given preference criteria, annotators who prefer a pattern such as slower motion are violating the instruction; their annotations then fall under labeler error.
>
> Although the paper is focused on the single criteria setup, SARA can be used for multi-preference criteria. For our problem setup, we assumed two subset categories for Transformer Encoder 2, preferred and non-preferred. With multiple criteria, such as “preferred-fast”, “preferred-slow”, “non-preferred”,  we would now have three categories which we train contrastively. However, we have not tested this due to lack of a multi-preference dataset for RL environments.
>
> **Weakness 2 and Question 2: SARA’s performance with fewer preference samples**
>
> Thank you for this question. We ran SARA and the Bradley-Terry based Preference Transformer (PT) using only *half* the originally given preference data. For these reduced preference datasets, we ran both 0% error rate and 20% error rate. At 0% error rate, we note SARA is less performant compared to PT on multiple datasets. However, at 20% error rate, SARA substantially outperforms PT on multiple datasets. Therefore, SARA still offers a robustness advantage over the Bradley-Terry based PT model in the low data regime.
>
> Below are mean normalized policy evaluation rewards (across 8 seeds), using **50% of preference datasets**. Values in bold are best (highest reward) in each row. The +- denotes standard deviation.
>
> **0% error rate**
>
> | Task | PT | SARA |
> | --- | --- | --- |
> | hop-medium-replay | 83.59 ± 14.3 | **85.00** ± 2.6 |
> | hop-medium-expert | **82.20** ± 33.8 | 53.90 ± 49.8 |
> | walk-medium-replay | 75.33 ± 4.0 | **76.98** ± 6.9 |
> | walk-medium-expert | **109.59** ± 1.3 | 107.68 ± 7.0 |
> | kitchen-partial | **59.96** ± 14.4 | 56.13 ± 22.0 |
> | kitchen-mixed | **54.38** ± 10.0 | 51.29 ± 10.8 |
>
> **20% error rate**
>
> | Task | PT | SARA |
> | --- | --- | --- |
> | hop-medium-replay | 54.99 ± 28.2 | **84.47** ± 3.1 |
> | hop-medium-expert | 63.00 ± 40.2 | **63.48** ± 43.1 |
> | walk-medium-replay | 71.55 ± 10.1 | **74.28** ± 10.8 |
> | walk-medium-expert | **108.56** ± 4.8 | 108.39 ± 4.7 |
> | kitchen-partial | 39.26 ± 26.6 | **41.95** ± 25.8 |
> | kitchen-mixed | 41.99 ± 16.3 | **48.16** ± 15.3 |

---

### Official Review · Reviewer_VMJU · 2025-11-01

**Soundness:** 3
**Presentation:** 3
**Contribution:** 3
**Rating:** 4
**Confidence:** 4

**Summary:**

This paper presents SARA, a new approach for preference-based reinforcement learning that aims to enhance robustness to noisy or inconsistent human feedback. SARA, instead of Bradley-Terry (BT) based methods, takes a set-based contrastive approach: it encodes preferred and non-preferred trajectories using a two-stage transformer encoder, producing a latent representation of preferences. The proposed method demonstrates strong performance on offline PbRL benchmarks, particularly when the quality of the human feedback is low.

**Strengths:**

- The paper addresses an important problem in PbRL literature, that the quality of the human feedback can vary.
- The proposed method outperforms the baselines when the label noise is introduced.

**Weaknesses:**

- The detailed design choices of the proposed method seem somewhat arbitrary. For instance, in Step 1 of Figure 1, the method splits the positive and negative embeddings into two subsets. However, it would also be possible to divide them into m subsets, where m is smaller than the batch size, or to omit the second encoder entirely and instead use a simple aggregate statistic—such as the mean or median—of the positive embeddings as z*_p. The authors should provide explicit justification for these design decisions, ideally supported by empirical comparisons demonstrating that the chosen configuration offers measurable advantages.
- The presentation of Figure 2 could be improved for interpretability. Each panel currently shows the performance of a single method while varying the label noise level. To facilitate direct comparison across methods, it would be clearer to reorganize the plots so that each panel corresponds to a fixed label quality, displaying the performance of all methods under that condition.
- In Table 1, the performance gap between the proposed method and the baselines does not seem significant except for hopper-med-replay and kitchen-mixed tasks when the error rate is 20%.
- The analysis accompanying Table 2 may be somewhat misleading. It is possible for multiple reward functions to induce the same optimal policy. In such cases, the learned reward function need not align closely with the original environment reward, and correlation with that reward does not necessarily indicate better policy quality. The authors should clarify this limitation and interpret the reported correlations accordingly.

**Questions:**

- SARA encoder takes as input multiple segments during training, but it seems the encoder takes as input a single segment during inference, according to Figure 1 step 3. Why is the input structure different for training and inference?
- Can the authors also evaluate the method on halfcheetah and Adroit?

---

> ### Author Response · Authors · 2025-11-21
> **Response**
>
> Thank you for your time in reading our paper and for your constructive response. We appreciate you noting our approach’s strong performance on offline PbRL benchmarks, particularly in the low quality data regime. Here we address your comments and questions in detail, and we would appreciate any further response.
>
> **Weakness 1: second encoder design choices**
>
> Thank you for these important points. We had addressed the choice of the number of subsets (which we refer to as a hyperparameter k) in lines 195-200 of page 4 and in Appendix A. We show that the model has low sensitivity to choice of k for low values of k (2, 3, and 4 subsets), and we show performance degradation at large k (k=16). At k=16, we have fewer trajectories per subset and we are closer to the limit of of k equal to batch size. At large k, model is unable to learn the discerning patterns for the full preferred set, ie the patterns distinguishing the preferred from the non-preferred set.
>
> You also noted we could omit the second encoder entirely and conduct contrastive learning between individual trajectories. We do this in two ablation experiments. Our paper already shows an ablation experiment in we first conduct contrastive learning between individual trajectories and then learn a Bradley-Terry model of reward (Table 4). We recopy these results here in the last column, Contrastive with BT.  In accordance with your suggestion, here we show a second ablation experiment called Contrastive with Mean $z_p{\*}$. We conduct contrastive learning between individual trajectories, and then we compute rewards $r_t=\text{cos}(z_t, z_p^{\*})$ using the mean of the preferred latents for $ z_p^{\*}$.
>
> The below tables show mean normalized policy evaluation rewards (8 seeds) with standard deviations. Values in bold are the highest per row.
>
> **10% error rates:**
>
> | Task | SARA | Contrastive with Mean $z_p^*$ | Contrastive with BT |
> | --- | --- | --- | --- |
> | hop-medium-replay | **83.66 (±3.5)** | 76.32 (±11.4) | 65.21 (±22.3) |
> | hop-medium-expert | 84.95 (±32.4) | **91.27 (±22.8)** | 80.65 (±29.4) |
> | walk-medium-replay | **78.18 (±7.9)** | 60.94 (±13.1) | 66.13 (±15.2) |
> | walk-medium-expert | 108.66 (±3.7) | 109.20 (±1.8) | **109.51 (±0.7)** |
>
> **20% error rate:**
>
> | Task | SARA | Contrastive with Mean $z_p^*$ | Contrastive with BT |
> | --- | --- | --- | --- |
> | hop-medium-replay | **82.94 (±5.8)** | 66.24 (±22.4) | 64.19 (±22.0) |
> | hop-medium-expert | 85.16 (±17.0) | **88.33 (±25.1)** | 86.93 (±21.8) |
> | walk-medium-replay | **76.29 (±13.2)** | 64.43 (±14.3) | 57.17 (±23.0) |
> | walk-medium-expert | 108.37 (±5.7) | 105.70 (±10.4) | **109.34 (±1.3)** |
>
> SARA shows a clear advantage over the ablations on the medium-replay datasets, achieving both higher mean returns and substantially lower variance. On the medium-expert datasets, SARA does not consistently outperform the ablations, but its performance remains competitive. We attribute this result to the composition of the medium-expert dataset: half of the trajectories originate from an expert policy. Even with noisy preference labels, contrastive learning on pairs where the expert and medium trajectories differ sharply can still recover a reasonable reward signal. In contrast, the medium-replay datasets contain more varying quality data, making them less robust to noise when learning on individual trajectories.

---

> ### Author Response · Authors · 2025-11-21
> **Response continued**
>
> **Weakness 2: Figure 2 interpretability**
>
> Thank you for noting interpretability issues in our Figure 2.  To clarify the intended message, we replaced Figure 2 with the following table in our revision.
>
> Our paper shows results for six labeling conditions (Tables 7-9): four error rates, script labeling by oracle, or exclusion of neutral queries. Here, we show the mean normalized episode evaluation returns across the six labeling conditions. The +- is the standard deviation across the six labeling conditions. We bold the **highest mean** and **lowest standard deviation** in each row. While baselines may win in mean or standard deviation as one-offs, SARA is the method that wins most frequently in mean and low variance across label conditions. In cases where SARA loses compared to baselines, the margin by which it loses is quite small. We believe this evidences SARA’s consistency in response to varying label quality.
>
> | Task | PT | PT+ADT | DPPO | SARA |
> | --- | --- | --- | --- | --- |
> | hopper-med-replay | 65.73 ± 25.49 | 71.29 ± 24.02 | 57.31 ± 26.71 | **81.49** ± **14.64** |
> | hopper-med-expert | 68.73 ± 36.17 | 77.09 ± **33.96** | 83.92 ± 36.05 | **85.22** ± 34.68 |
> | walker2d-med-replay | 72.35 ± 12.65 | 72.58 ± 16.25 | 43.93 ± 28.13 | **75.95** ± **11.29** |
> | walker2d-med-expert | 107.10 ± 9.41 | 106.42 ± 10.82 | **108.73** ± **0.42** | 108.70 ± 4.57 |
> | kitchen-partial | 59.69 ± 17.90 | 60.76 ± **17.06** | 38.50 ± 18.51 | **61.01** ± 17.93 |
> | kitchen-mixed | **49.85** ± 15.53 | 48.57 ± 16.21 | 45.68 ± 17.97 | 49.03 ± **11.40** |
>
> Note that within a labeling condition, there is variance arising due to the 8 seeds. We account for both sources of variance (across the labeling condition and within labeling condition) via the law of total variance.
>
> Our paper also provides comparison across methods for each fixed label type in Tables 7-9.
>
> **Weakness 3: Performance gap against baselines**
>
> We agree that SARA does not beat every baseline at every label condition.  Rather, SARA is a reliable model, compared to baselines, because it displays less variability in response to label condition. Baseline methods may beat SARA on a particular label condition and dataset, but those same methods suffer in response to increasing error rate. We believe the above table supports this message.
>
> SARA also shows strong significantly stronger correlation to the environmental rewards at all error rates (Table 6), and we infer that implies stronger alignment to the preference criteria than baselines. We expound on this result and the interpretation in response to the following weakness.

---

> ### Author Response · Authors · 2025-11-21
> **Response continued**
>
> **Weakness 4: Interpretation of Pearson correlation to environment rewards**
>
> Thank you for your observation that the learned reward function need not align closely with the original environment rewards. We note that previous PbRL works also used this as a metric for evaluating their learned reward models (Choi et al., 2024; Zhang et al., 2024; Liu et al., 2025)  . We provide the following justification for this metric.
>
> We use the correlation to the environment reward function as a proxy for the adherence to the preference criteria. We can do this because the preference criteria given to the human labelers does in fact align closely to the design of the environment rewards.
>
>  For example, the instruction given to the human teacher on the Hopper datasets is:
>
> The hopper robot aims to move to the right as far as possible while minimizing
> energy costs. If the hopper robot lands unsteadily, lower your priority even if the distance
> traveled moved during a segment is longer than the other. If the two robots are almost tied on
> this metric, choose the segment by the distance that the robot has moved.” (Preference Transformer, Appendix C).
>
> Likewise the environment reward function provides rewards at each timestep in proportion to the velocity of its movement and remaining in a stable upright position. Please see for the detailed reward function:
>
> https://gymnasium.farama.org/environments/mujoco/hopper/. Thus, human preference criteria and the environment reward function incentivize the same behavior. Therefore, we can use alignment to the per-step environment reward function as a proxy measure for alignment to the preference criteria.
>
> We agree that better correlation does not result in a better policy. Our paper states: “SARA demonstrates substantially better correlation with environment rewards compared to baselines across most datasets, though we find that this correlation advantage not always translate directly to policy performance improvements.” The subsequent paragraphs then noted instances in which the baselines achieved better policy episode returns despite lower correlation to ground truth.
>
> In summary, PbRL research aims to capture human preferences, not necessarily learn a policy that achieves higher episode returns as measured by the environment. Past works primarily focused only on the latter as a metric. However, we validated our model using both this traditional metric and the Pearson correlation to the environment per-step reward, where the latter is a proxy for the preference criteria.
>
> We agree that that these points are subtle and difficult to discern. In our revision, we further clarified that these are two distinct metrics of assessing adherence to preference criteria, as well as the limitations of each metric.
>
> **Question 1: Training vs Inference**
>
> Please note that the training in step 1 and the $z_p^*$ inference in step 2 use only the preference dataset. At step 3, we are done with the preference dataset, and we want to compute rewards in an RL algorithm. If we are doing offline RL, then we have a large buffer of sampled trajectories. In this case we would indeed pass all the N trajectories through the encoder to get a $z_t^n, n \in N,$  for each timestep $t$ of each trajectory n. However, if we are doing online RL, we might not have a large buffer of offline trajectories. Rather, we sample one step at a time, and we would compute the $r_t$ at each step online. We showed just a single trajectory in in step 3 in order to avoid making any assumptions on the RL training paradigm (offline or online, replay buffer or no buffer).
>
> Also we would further like to emphasize a subtle point. Step 1 learns to encode **sets** of trajectories and step 2 conducts inference on the **set** of preferred trajectories. At step 3 we are doing reward inference for RL, in which we sample trajectories from any arbitrary policy. Here we no longer have preference labels for the sampled trajectories. In step 3, we are encoding **individual** trajectories to get a reward estimate at each timestep for each individual trajectory.

---

> ### Author Response · Authors · 2025-11-21
> **Response continued**
>
> **Question 2: Halfcheetah and Adroit**
>
> The Adroit results are provided in Appendix B.3, Tables 10-13. We relegated these results to the Appendix because the returns are similar across all models and also very high variance. The IQL results using the oracle are likewise very high variance.
>
> We ran the model on Halfcheetah using preference datasets provided by An et al. Below are the the average normalized policy evaluation rewards (across 8 seeds), using human-labeled preference data. Highest values and those within 1% are bold. The +- denotes standard deviation.
>
>  **0% error rate**
>
> | Task | PT | PT+ADT | DPPO | SARA |
> | --- | --- | --- | --- | --- |
> | halfcheetah-medium-replay | 40.94 ± 2.7 | **42.85 ± 1.7** | 39.94 ± 4.3 | 41.65 ± 2.0 |
> | halfcheetah-medium-expert | 86.62 ± 14.2 | 89.46 ± 9.4 | **92.18 ± 8.5** | 86.56 ± 13.5 |
>
> **20% error rate**
>
> | Task | PT | PT+ADT | DPPO | SARA |
> | --- | --- | --- | --- | --- |
> | halfcheetah-medium-replay | 41.16 ± 1.9 | **42.25 ± 2.2** | 38.59 ± 6.9 | **42.08 ± 2.1** |
> | halfcheetah-medium-expert | 87.97 ± 11.1 | 89.18 ± 10.4 | **92.32 ± 7.6** | 88.41 ± 10.4 |
>
> The models perform similarly and SARA provides competitive results against baselines.

---

### Author Response · Authors · 2025-12-03
**Rebuttal Summary**

We thank the reviewers for their detailed feedback. Multiple reviewers noted our method’s strong empirical performance against baselines in the real world problem of low quality human feedback (VMJU and LfAy). Reviewers LfAy and upAD appreciated our method’s simplicity, rendering it practical in real settings. Reviewer LfAy mentioned SARA’s versatility to cross-task preference transfer and reward shaping. Reviewer y4XU noted our “extensive experiments across multiple environments, demonstrating its effectiveness”. We appreciate the reviewers rated our presentation, soundness, and contribution as good (VMJU and LfAy).

The reviewers requested clarity on the intended message of a results figure and the interpretation of the Pearson correlation evaluation metric. Reviewers also requested clarity on motivation for our model’s design choices, in particular the cosine similarity reward and the set-based encoding of preference. Some reviewers asked for further empirical results. We responded to all the reviewers thoroughly to address their concerns, and we have updated our paper in accordance. Here we provide a global list of the major changes in response to the reviewers.

**Experimental**

1. We apologize that the intended message of Figure 2 in our original manuscript was unclear (reviewers VMJU and upAd). We replaced this Figure 2 with Table 2 along with a detailed caption. This table shows mean and standard deviation of model performance across six preference labeling conditions: human labeled with four labeling error rates, excluding neutral preferences, and script labeled. While baselines may win in mean or standard deviation as one-offs, SARA is the method that wins most frequently in highest mean and lowest variance across label conditions. In cases where SARA loses compared to baselines, the margin by which it loses is quite small. Thus, SARA displays the most consistent results as labeling condition varies, and it does not exhibit the large fluctuations across labeling conditions that we see in the baseline models.
2. Reviewer VMJU found our interpretation of the Pearson correlations to environmental rewards as misleading. We added a detailed Evaluation Metrics paragraph (page 6) to explain our metrics and the merits of the Pearson correlation to environmental rewards. The preference criteria given to labelers aligns closely with the true environment given reward function. Thus, correlation to the true environment rewards serves as a proxy measure for alignment to the preference criteria. This metric was also used in previous works, which we cite in this paragraph on page 6. We agree with the reviewer that this correlation does not always result in a better policy; as such we explain this point in lines 410-418 on page 8 of the paper. We note also that the goal of PbRL research is to align with preferences,  which may not necessarily result in a more performant policy. For this reason, the Pearson correlation is a complementary and insightful metric.
3. Reviewers VMJU and upAd requested empirical evaluation on additional datasets. We ran our model and baselines on HalfCheetah medium-replay and medium-expert datasets, with and without labeler error. We updated our Table 1 with additional results, showing SARA performs competitively against baselines on these additional datasets.
4. In response to reviewer LfAY, we conducted preference data scaling experiments (Appendix D). We show that in the low data regime (50% of preference data) with label noise (20% error rate), SARA provides a substantial robustness advantage against the Bradley-Terry based PT model. However, our experiment shows SARA is less performant against PT in the low data regime with 0% error.

**Theoretical:**

1. Reviewer upAd requested a clearer explanation for our model’s robustness compared to BT models. We re-worded the text on page 9 (Section 5 Mechanistic Analysis).
2. Reviewer y4XU asked why we use the cosine similarity as the reward. We had previously included our theoretical motivation for this reward in the Appendix. To improve clarity, we moved the explanation into the main paper with paragraph “SARA reward: theoretical justification” on page 5. Here we explain that the cosine similarity reward arises naturally from the SimCLR contrastive loss objective, which is now provided on page 4.
3. Reviewers LfAy and upAd asked whether the SARA model can be used when there are heterogenous preference criteria. The problem scope for our work and the baseline models is a single specific preference criteria per task. Thus, labeling in accordance to a different preference criteria than the one given falls under the umbrella of labeler error. However, we update our Limitations and Future Work section (page 10) to explain that SARA can be straight-forwardly applied to heterogenous preference criteria. We are currently unaware of RL preference datasets to test this on, but this is an exciting future application of SARA.

---

### Meta-Review · Area_Chair_UJfY · 2025-12-16

**Summary:**

While the paper proposes a SARA method, the experimental scope is narrow and lacks new controls; the legend is disorganized, and key issues such as heterogeneous bias among multiple annotators, multimodal trajectories, noise robustness, and failure boundaries are not fully explained or verified. SARA relies on a fixed prototype, and adding or biasing preferences requires global retraining. Furthermore, the sample size-performance relationship is not explained, and its performance with small samples and relative data efficiency compared to Bit-Proof are unknown. It is urgent to explain how the prototype is updated online and whether it is more data-intensive in scenarios with few samples.

**Reviewer Concerns:**

Although the authors have supplemented their responses with experiments, ablation methods, and baseline comparisons, the rationality and effectiveness of the proposed method have not been completely resolved.

**Reviewer Scores:**

All three reviewers maintained negative scores.

---

### Decision · Program_Chairs · 2026-01-26

Reject